# PerturBench: Benchmarking Machine Learning Models for Cellular Perturbation Analysis

**Yan Wu[1]***, **Esther Wershof[1]***, **Sebastian M Schmon***[†], **Marcel Nassar[1]***, **Błażej Osiński***[†],
**Ridvan Eksi[1]***, **Zichao Yan[1]***, **Rory Stark[1]**, **Kun Zhang[1]**, **Thore Graepel[2]**[†][‡]

[1]Altos Labs    [2]University College London

## Abstract

We introduce a comprehensive framework for modeling single cell transcriptomic responses to perturbations, aimed at standardizing benchmarking in this rapidly evolving field. Our approach includes a modular and user-friendly model development and evaluation platform, a collection of diverse perturbational datasets, and a set of metrics designed to fairly compare models and dissect their performance. Through extensive evaluation of both published and baseline models across diverse datasets, we highlight the limitations of widely used models, such as mode collapse. We also demonstrate the importance of rank metrics which complement traditional model fit measures, such as RMSE, for validating model effectiveness. Notably, our results show that while no single model architecture clearly outperforms others, simpler architectures are generally competitive and scale well with larger datasets. Overall, this benchmarking exercise sets new standards for model evaluation, supports robust model development, and furthers the use of these models to simulate genetic and chemical screens for therapeutic discovery.

## 1 Introduction

Perturbing biological systems, such as cells, using small molecules and genetic modifications can help researchers to uncover causal drivers of diseases and identify potential therapeutic targets [9, 26, 11]. Advances in CRISPR technology and lab automation have enabled these experiments, which we refer to as perturbation screens, to be conducted at scale with up to hundreds of thousands of perturbations applied in parallel in a single experiment [49]. These perturbation screens have been combined with modern RNA-sequencing technology to measure gene expression profiles at single cell resolution, creating atlases of cellular snapshots that reveal perturbation effects [35, 54, 36, 1, 15, 9, 51, 61].

However, measuring the perturbation effects of all roughly $20,000$ protein coding genes or $10^{60}$ drug-like chemicals remains prohibitively expensive, especially when taking into account combinations of perturbations and different tissues, cell types, and cell lines [43, 44]. As a result, there has been a growing interest in generative machine learning approaches that can predict the effects of perturbations on gene expression.

Specifically, researchers have developed models that can generate counterfactual, out of sample (oos) predictions of perturbation effects [19]. One use case, which we call *covariate transfer*, involves training a model on perturbation effects measured in a set of covariates (e.g., cell lines) and predicting those effects in another covariate where the perturbation-covariate pairs have not been observed.

---

*Equal contribution. Correspondence to ywu@altoslabs.com.

[†]Work done while at Altos Labs

[‡]{ewershof, mnassar, reksi, zyan, rstark, kzhang}@altoslabs.com
sebastian.schmon@gmail.com, b.osinski@mimuw.edu.pl, t.graepel@ucl.ac.uk

*combo prediction* involves training a model on individual perturbation effects and predicting the effects of multiple perturbations in combination. The ultimate goal is to enable in-silico screens across the vast space of unobserved perturbations to accelerate therapeutic discovery.

**Related Works**    Comparing the performance of published models has been challenging due to inconsistent benchmarks with different datasets and metrics. The sc-perturb database provides datasets with unified metadata, but does not benchmark models [40]. The NeurIPS 2023 perturbation prediction challenge [53] was a major achievement in standardizing benchmarks, providing a novel chemical perturbation dataset with scRNA-seq readouts measured in PBMCs. The challenge used the covariate transfer task, with metrics including mean RMSE, MAE, and cosine similarity of predicted vs ground truth log p-values. The challenge attracted a large number of submissions, many of which were inspired by published models such as chemCPA [53].

Ahlmann-Eltze et al. [3], Wenteler et al. [56], Csendes et al. [12], and Wong et al. [58] evaluate single-cell foundation models (scFM) such as scGPT, scFoundation, scBERT, Geneformer and UCE, in the context of perturbation response modeling. These studies focus on how these general-purpose models can be fine-tuned for this task, using task-specific models such as GEARS, CPA as well as other baselines (e.g., mean prediction, kNN, random forest, linear models) to highlight the limitations of scFMs for this task. Notably, Wong et al. [58] used our baseline models and rank metric to establish the performance of scGPT and GEARS. These works mostly use well-known model fit metrics such as RMSE/MSE and Pearson correlation between averaged predicted and ground truth expression profiles (Pearson Delta, Pearson LogFC). Wenteler et al. [56] also proposed various distributional metrics, including E-distance, which is equivalent to our energy distance based MMD metric.

Kernfeld et al. [27] systematically assessed a wide array of perturbation response prediction models, with a focus on models that use gene regulatory networks as a form of prior knowledge. A central finding of their work was that simple baselines often matched or outperformed more sophisticated models such as GEARS and Geneformer, which confirm the robust performance of simple approaches in this domain. Recent works by Li et al. [30] and Li et al. [31] provide more comprehensive benchmarks of a large set of deep learning models across diverse datasets and metrics. Beyond conventional evaluation, their work introduces novel tasks, such as unseen perturbation/covariate transfer [30, 31] and cell state transition prediction [30]. Notably, while scFMs can excel on unseen perturbation prediction, simpler models often show better performance in the unseen covariate prediction [30].

**Contributions**    In this work, we **(1)** introduce a highly modular and user-friendly framework in the form of a codebase (Perturbench) for model development and evaluation, **(2)** curate diverse perturbational datasets and define biologically relevant tasks, **(3)** develop metrics that enable rigorous model comparison and capture key failure modes, and **(4)** perform extensive evaluation of published perturbation models, strong baselines, and individual model components. Figure 1 illustrates our approach.

We reproduce key components of published models that cover a spectrum of architectures, and evaluate them alongside strong baseline models. We specifically test the models on difficult tasks, simulating how they will be deployed in real-life contexts. Our findings reveal that some widely used models are prone to "mode" or "posterior" collapse (see Appendix C.7 for more details). Since a common use-case of these models is to run in-silico screens that rank perturbations by a desired effect (e.g. reversing a disease state) [55], we propose rank metrics complementary to traditional measures of model fit (e.g. root mean squared error (RMSE)) that specifically assess the models' ability to accurately order perturbations and detect model collapse. In addition, we demonstrate that models with simple architectures can outperform more sophisticated models when trained on larger datasets.

We anticipate that our codebase, together with this benchmark and the accompanying metrics, will serve as a valuable framework for the community to develop more robust perturbation response prediction models. The PerturBench library can be found on GitHub at `https://github.com/altoslabs/perturbench/`.

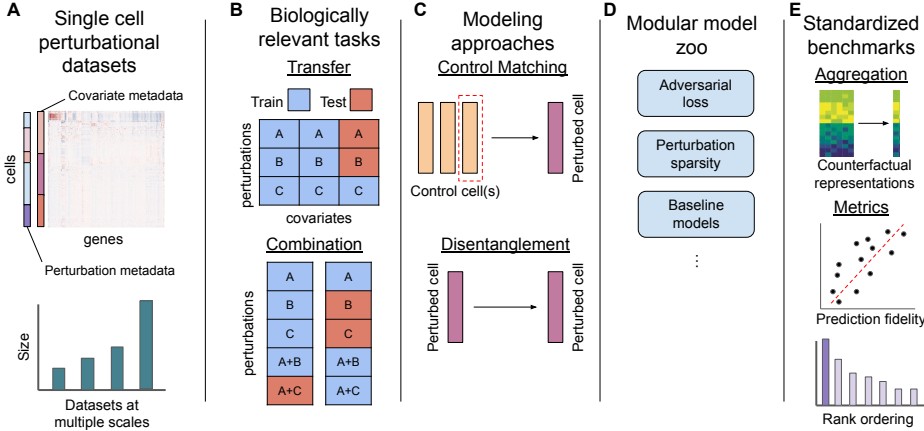

Figure 1: **A**) Single cell perturbational datasets at multiple scales. **B**) Biologically relevant covariate transfer and combinatorial prediction data splits. **C**) Dataloaders support two training strategies: 1) control matching which involves mapping a control cell to a perturbed cell and 2) disentanglement which maps a perturbed cell to itself. **D**) A model zoo with modular components such as relevant baseline models, adversarial loss, perturbation sparsity, and others. **E**) Standardized benchmarking suite supporting flexible pipelines and metrics for evaluating models

## 2 Datasets and Tasks

Many published models have been evaluated on relatively small datasets, where most of the data are seen in the train split. However, in real-world settings, we often have complex datasets that only contain a small fraction of the perturbation effects we are interested in predicting. Thus, we select datasets and create tasks that mirror these real-world challenges. We select six published datasets, Norman et al. [38], Srivatsan et al. [51], Frangieh et al. [18], McFaline-Figueroa et al. [37], Jiang et al. [23], and Szałata et al. [53] (OP3) which include at least 100 perturbations and cover a diversity of perturbation modalities (chemical vs genetic), combinatorial perturbations, dataset sizes, and covariates. We provide a cursory overview in Table 1 and more information in Appendix D.1. Here, we define a biological state as a unique set of covariates that we plan to model (e.g. cell type/line). Dataset preprocessing details can be found in Appendix D.3.

Table 1: Summary of benchmarking datasets.

| Dataset | Single pert. | Dual pert. | Modality | Primary cells | Biological states | Cells | Task |
|---|---|---|---|---|---|---|---|
| Srivatsan20 | 188 | 0 | chemical | ✗ | 3 | 178,213 | *covariate transfer* |
| Frangieh21 | 248 | 0 | genetic | ✗ | 3 | 218,331 | *covariate transfer* |
| Jiang24 | 219 | 0 | genetic | ✗ | 30 | 1,628,476 | *covariate transfer* |
| McFalineFigueroa23 | 525 | 0 | genetic | ✗ | 15 | 892,800 | *covariate transfer* |
| Norman19 | 155 | 131 | genetic | ✗ | 1 | 91,168 | *combo prediction* |
| OP3 | 144 | 0 | chemical | ✓ | 4 | 296,147 | *covariate transfer* |

We create *covariate transfer* tasks for the Srivatsan20, Frangieh21, Jiang24 and OP3 datasets as well as a *combo prediction* task for the Norman19 dataset. In addition, we study two **scenarios**: *data scaling* and *imbalanced data*. In the former, we benchmark model performance with increasing training data. In the latter, we simulate increasing the imbalance of perturbations observed in different covariates. Details of the experiments are in sections 5.3 and C.6 respectively. The aim of both scenarios is to simulate how models will be deployed in practice, where there are often complex covariates, imbalanced datasets, and/or large amounts of missing data [see e.g. 16]. Additional details about data splitting implementation can be found in the Appendix D.4.

# 3 Perturbation Prediction Models

## 3.1 Modeling counterfactuals

Perturbation response models aim to predict out-of-sample effects of genetic or chemical interventions on cells. Here, we define out of sample as predicting effects in unobserved covariates or unobserved perturbation combinations. However, RNA sequencing technology destroys the cell, making it impossible to observe its gene expression state before and after perturbation. Published models use two main strategies to learn representations of perturbation effects: *matching methods* to match control and perturbed cells, or *disentanglement* strategies within autoencoder architectures to separate the effects of perturbations from the baseline cell state.

**Matched Controls**  Matching treated outcomes to controls is a common approach to identify treatment effects [see e.g. 52, Section 1.3 for a historical summary]. In the context of perturbation effect prediction, matching control and perturbed cells has been used by a variety of published models such as GEARS [46], scGPT [13], and scFoundation [20]. For the matching approach, ensuring that the control cell is from the same cell type/line, experiment or batch as the perturbed cells helps reduce potential confounding effects, but cannot account for unobserved sources of variance. A more complex approach is to use optimal transport to identify the control cell most likely to transition into a given perturbed cell [24, 10, 28]. This enables prediction of the full distribution of cellular responses, instead of just the average response.

**Disentanglement**  An alternative to *matching methods* involves *disentanglement* [6], which enables models to separate the unperturbed cellular state and the perturbation effect. The compositional perturbation autoencoder (CPA) [34] uses an adversarial classifier to ensure that the unperturbed "basal" state is free of any perturbational information, forcing the perturbation encoder to learn a meaningful representation of the perturbation. These representations are added to control cell encodings during inference to generate counterfactual predictions. Biolord [41] partitions the latent space into subspaces and optimizes those latent spaces to represent covariates and perturbations, which can be recombined during inference to generate counterfactual predictions. sVAE [33] leverages recent results by Lachapelle et al. [29] demonstrating that enforcing a *sparsity* constraint can induce disentanglement. Bereket and Karaletsos [7] build on sVAE using an additive conditioning for the perturbations, leading to SAMS-VAE which has biologically interpretable latent encodings.

## 3.2 Models for benchmarking

In this paper, we implement a range of perturbation response models with diverse architectures such as CPA, Biolord, SAMS-VAE and GEARS. Our aim is to assess the core modeling assumption behind these models, such as the adversarial classifier in CPA and the sparse additive mechanism in SAMS-VAE. See Appendix D.5 for details. These models are marked with $^*$ following the model name (e.g. CPA$^*$).

To investigate the effect of disentanglement, we ablate the adversarial component from CPA and refer to the new version as CPA$^*$ (noAdv). The rest of the model is unchanged. Despite the basal state of cells (i.e. $z_{basal}$) are contaminated with perturbation and covariate information, counterfactual gene expressions are still generated by adding the target perturbation embedding to $z_{basal}$. We verified that our CPA implementation performs at least as well as the published version in C.7.4.

Working under the same hypothesis, we removed the binary mask from SAMS-VAE$^*$, thus discarding the sparsity assumption which drives the disentanglement and offers interpretability. We also discard the distributional assumption on the perturbation embedding, arriving at a simplified version which we refer to as SAMS-VAE$^*$ (S) that is free of any global latent variables.

We also experiment with scGPT, a single-cell gene expression foundation model [13], to embed gene expression — as inputs to CPA and our Latent Additive baseline model. Our aim is to understand whether using cell embeddings from a pretrained foundation model improve performance.

## 3.3 Baseline models

We also implement and benchmark the following baselines:

**Linear** The linear baseline model uses the *control matching* approach. Given a perturbed cell, $x'$, we sample a random control cell with *matched* covariates, $x$, and reconstruct $x'$ by applying one linear layer given the perturbation and covariates:

$$x' = x + f_{\text{linear}}(p_{\text{one\_hot}}, cov_{\text{one\_hot}}), \tag{1}$$

where $p_{\text{one\_hot}}$ denotes the one-hot encoding of the perturbation and $cov_{\text{one\_hot}}$ denotes one-hot encodings of covariates (e.g. cell type/line).

**Latent Additive** We extend the linear model into a stronger baseline which we call Latent Additive, by encoding expression values and perturbations into a latent space $\mathsf{Z} \subseteq \mathbb{R}^{d_z}$, i.e.

$$z_{\text{ctrl}} = f_{\text{ctrl}}(x), \quad \text{and} \quad z_{\text{pert}} = f_{\text{pert}}(p_{\text{one\_hot}}),$$

Subsequently, we reconstruct the expression value by decoding the added latent space representation $x' = f_{\text{dec}}(z_{\text{ctrl}} + z_{\text{pert}})$. All functions $f_{\text{dec}}, f_{\text{ctrl}}, f_{\text{pert}}$ are implemented as multilayer perceptrons (MLPs) with dropout [50] and layer normalization [5].

**Decoder Only** We introduce another class of baseline which we refer to as *Decoder-Only*, that does not leverage gene expression and aims to predict the perturbation effect solely from covariates, or perturbation, or a mix of both. Consequently, model prediction can be expressed as $x' = f_{\text{dec}}(z)$ for $z \in \{p_{\text{one\_hot}}\} \cup \{cov_{\text{one\_hot}}\} \cup \{(p_{\text{one\_hot}}, cov_{\text{one\_hot}})\}$. This baseline can be used to establish a performance lower-bound when it does not receive any perturbation information, simulating mode or posterior collapse where the model predicts the same effect for every perturbation.

## 4 Benchmarking

Existing perturbation response modeling studies use different metrics, making direct comparison between models difficult. Here, we develop a standardized, modular benchmarking suite with a variety of metrics that mimic key downstream applications and capture common model failure modes. See Appendix A for implementation details.

### 4.1 Population Aggregation

Since we only have conditionally paired populations of control cells and perturbed cells, a model has to take unperturbed (i.e. control) cell states and predicts what their states *would* look like if they had been perturbed. To evaluate these counterfactual predictions, we thus compute aggregate measures from the control and perturbed cell populations, and compare the predicted aggregates to the ground truth aggregates. We use the population mean and log fold-change (LogFC) to assess the average response to perturbation, and compute distributional metrics such as maximum mean discrepancy (MMD) and differential gene expression (DEG) t-scores to assess the full distribution of responses.

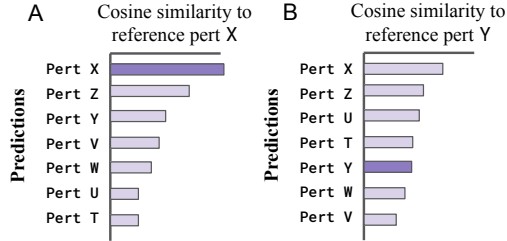

Figure 2: Visualization of the ranking approach. We measure which perturbation in the data is closest to the predicted perturbation as measured by the closeness of their transcriptomes. In case A the rank metric for prediction X is $\text{rank}(X) = \frac{0}{6} = 0$, in case B $\text{rank}(Y) = \frac{4}{6} = 0.67$.

### 4.2 Metrics

We select a range of metrics that evaluate different aspects of the predicted perturbation response accuracy. We use RMSE (as recommended in Ji et al. 22) to compare the average predicted cell states to the observed cell states and cosine similarity to assess the fidelity of predicted versus observed effects (LogFCs). However, as we show in Appendix C.7, these "global" fit-based metrics fail to fully capture all aspects of a model's performance. Hence, we introduce a set of rank-based metrics that can be seen as a measure of *specificity* of the model to different perturbations.

### 4.2.1 Rank Metrics

Since the space of possible genetic or chemical perturbations is massive, a common application of these models is to rank perturbations by by their ability to induce some desired cell state, and select only the top perturbations for experimental validation [55].

Unfortunately, there are no metrics developed for perturbation ordering, which results in models that generate predictions with high cosine similarity or low RMSE to the observed gene expression, but fail to capture smaller but key changes that uniquely distinguish the effects of one perturbation from others. In particular, mode or posterior collapse where a model always generates the same prediction irrespective of target perturbation may still result in decent cosine similarity or RMSE on particular dataset.

We find it difficult to use existing information retrieval metrics such as the mean reciprocal rank [42] as that would require a specific desired cell state to create rankings, potentially biasing our evaluation. Therefore, we introduce a rank-based metric that measures, for a given observed perturbation, how close the model prediction is to the observation, compared to predictions made for other perturbations. The rank metric is computed on a per perturbation basis:

$$\text{rank}_{\text{average}} := \frac{1}{p} \sum_{i=1}^{p} \text{rank}(\hat{x}_i), \quad \text{rank}(\hat{x}_i) := \frac{1}{p-1} \sum_{\substack{1 \le j \le p \\ j \ne i}} \mathbb{I}(\text{dist}(\hat{x}_j, x_i) \le \text{dist}(\hat{x}_i, x_i)), \quad (2)$$

where $p$ is the number of perturbations that are being modelled, $\hat{x}_i, x_i$ are the predicted and observed (average) expression value of perturbation $i$, and dist is a generic distance. Figure 2 shows two examples of perturbations predictions and the respective computations of rank metrics. The rank metric is always a number between 0 and 1, where 0 is a perfect score and 0.5 is the expected score of a random prediction. Each fit-based metric has a corresponding rank metric. We verified the calibration of this rank metric in Appendix C.7.5.

### 4.2.2 Distributional Metrics

We evaluate model predictions with metrics that capture the full distribution of perturbation response across cells, such as MMD and differentially expressed gene (DEG) recall.

Our MMD metric computes the energy distance between predicted and the ground truth perturbed cells, either in the gene expression or principal component analysis (PCA) space. To embed cells with PCA, we fit a PCA model to the ground truth test split, retaining the top 256 PCs, and project predicted cells onto the ground truth PCs. DEG recall measures the fraction of ground truth DEGs that can be recalled from the predicted cells. DEGs for the predicted and ground truth perturbed cells are computed relative to the control cells, using the scanpy's `tl.rank_genes_groups` method with default parameters [57]. The t-scores are sorted and the top 20 are retained for comparison. We leave DecoderOnly DEG recall empty as it cannot model variance in perturbation response.

### 4.3 Benchmarking Rules

Since model performance often varies with hyperparameters, we run hyperparameter optimization (HPO) for every model in each dataset, task and scenario to ensure accurate model comparisons. Specifically, we use `optuna` [4] with the default tree-structured Parzen estimator [8]. Each HPO run includes at least 60 trials with 6 trials running in parallel, and we select the best hyperparameters using a combination of the RMSE and RMSE rank metric, i.e. $\text{RMSE} + 0.1 \cdot \text{rank}_{\text{RMSE}}$. We find this approach results in good overall performance. Additional details are in Appendix D.6, along with the best hyperparameters for each model/dataset/task. For the best hyperparameter configuration, we run model training four additional times with different seeds to assess stability.

## 5 Experiments

In this section, we summarize the results of the *covariate transfer* and *combo prediction* **tasks** using different datasets, as well as the *data scaling* scenario. Additional results, including results for the Frangieh21 and OP3 dataset, *imbalanced data* scenario, and implementation details can be found in Appendices C, D, and D.7.

Table 2: Results of the *covariate transfer* experiment measuring generalization across cell lines in the `Srivatsan20` dataset. Mean ± one standard deviation reported across 5 seeds. Best performance per metric is indicated in bold. The same convention applies to subsequent tables.

| Model | Cosine LogFC | Cosine LogFC rank | MMD PCA | DEG recall |
|---|---|---|---|---|
| CPA$^*$ | $0.38 \pm 6 \times 10^{-3}$ | $0.15 \pm 1 \times 10^{-2}$ | $0.53 \pm 4 \times 10^{-3}$ | $0.007 \pm 2 \times 10^{-3}$ |
| CPA$^*$ (noAdv) | $0.40 \pm 5 \times 10^{-3}$ | $\mathbf{0.09 \pm 4 \times 10^{-3}}$ | $\mathbf{0.49 \pm 1 \times 10^{-2}}$ | $0.004 \pm 2 \times 10^{-3}$ |
| CPA$^*$ (scGPT) | $0.39 \pm 9 \times 10^{-3}$ | $0.13 \pm 2 \times 10^{-2}$ | - | - |
| SAMS-VAE$^*$ | $0.44 \pm 1 \times 10^{-3}$ | $0.17 \pm 1 \times 10^{-2}$ | $0.69 \pm 1 \times 10^{-2}$ | $0.000 \pm 1 \times 10^{-4}$ |
| SAMS-VAE$^*$ (S) | $\mathbf{0.53 \pm 1 \times 10^{-2}}$ | $0.12 \pm 2 \times 10^{-2}$ | $0.79 \pm 1 \times 10^{-2}$ | $0.000 \pm 0$ |
| Biolord$^*$ | $0.18 \pm 1 \times 10^{-1}$ | $0.37 \pm 2 \times 10^{-2}$ | $4.3 \pm 4 \times 10^{0}$ | $0.000 \pm 1 \times 10^{-4}$ |
| LA | $0.45 \pm 2 \times 10^{-3}$ | $0.13 \pm 4 \times 10^{-3}$ | $2.0 \pm 2 \times 10^{-1}$ | $0.000 \pm 0$ |
| LA (scGPT) | $0.50 \pm 4 \times 10^{-3}$ | $0.13 \pm 7 \times 10^{-3}$ | - | - |
| Decoder | $0.35 \pm 5 \times 10^{-3}$ | $0.16 \pm 1 \times 10^{-2}$ | $1.9 \pm 5 \times 10^{-3}$ | - |
| Decoder (Cov) | $0.30 \pm 1 \times 10^{-2}$ | $0.47 \pm 9 \times 10^{-3}$ | - | - |
| Linear | $0.16 \pm 1 \times 10^{-2}$ | $0.28 \pm 5 \times 10^{-3}$ | $0.76 \pm 9 \times 10^{-4}$ | $0.004 \pm 3 \times 10^{-4}$ |

## 5.1 Predicting Perturbation Effects Across Cell Lines

We begin with the *covariate transfer* task and assess each model's ability to predict the effects of drug treatment in cell lines where the drugs have not been observed. For each cell line in the `Srivatsan20` dataset, we held out 30% of the perturbations for validation and testing, ensuring that any held-out perturbations have been observed in the two other cell lines. The results are summarized in Table 2. Of the three published models, CPA$^*$ performs best on the rank metrics, while SAMS-VAE$^*$ is slightly worse. However, SAMS-VAE$^*$ performs better on the fit based RMSE and cosine LogFC metrics. BioLord$^*$ underperforms both models on all metrics.

We performed ablation studies by removing the adversarial component from CPA$^*$, resulting in the CPA$^*$ (noAdv). Strikingly, CPA$^*$ (noAdv) outperforms the original version and is the best performing model overall on the rank metrics. We also ablated the sparse mask and global latent variables from SAMS-VAE$^*$, resulting in SAMS-VAE$^*$ (S), and observe that SAMS-VAE$^*$ (S) also beats the original implementation on all metrics. These results highlight the need for a modular model development platform that enables ablation studies.

Our Decoder-Only baseline which does not leverage perturbation information, Decoder (Cov), also performs well on cosine LogFC and RMSE fit, yet does no better than random on rank metrics. This suggests that Decoder (Cov) can find a single expression vector that achieves a decent fit to all perturbations in a given cell line, highlighting the need for our rank metrics that assess whether models can correctly order perturbations. Appendix C.7 contains a more detailed assessment, where we are able to establish that SAMS-VAE$^*$ (S) as well as our baselines, are less prone to mode collapse, based on analysis in the `Srivatsan20` and `Frangieh21` datasets.

By replacing gene expression inputs with scGPT cell embeddings, we are able to see marginal improvement in the performance for CPA$^*$ and LA.

We see large differences in model performance on the MMD metric computed in the PCA space (MMD PCA). Models using autoencoder architectures (CPA and SAMS-VAE), tend to have better performance than our baseline models (Latent Additive, Decoder Only). All models performed poorly on our differentially expressed gene recall (DEG recall) metric. Additional metrics for `Srivatsan20` can be found in Table 5.

Since the `Srivatsan20` dataset contained mostly cancer cell lines, we also benchmarked models on the `Frangieh21` and `OP3` datasets, which contain melanoma cells in co-culture with immune cells and primary peripheral blood monocyte cells (PBMCs) respectively. With the `Frangieh21` dataset, we assess models' ability to transfer genetic perturbation effects from simpler cell systems to a more complex co-culture system, finding similar trends in model performance, with more details in Appendix C.3.

Table 3: Results of the *combo prediction* experiment. Model performance predicting dual perturbation effects in the `Norman19` dataset.

| Model | Cosine logFC | Cosine LogFC rank | MMD PCA | DEG recall |
|---|---|---|---|---|
| CPA$^*$ | $0.76 \pm 4 \times 10^{-3}$ | $0.0072 \pm 2 \times 10^{-3}$ | $2.2 \pm 2 \times 10^{-2}$ | $0.032 \pm 4 \times 10^{-3}$ |
| CPA$^*$ (noAdv) | $0.77 \pm 1 \times 10^{-2}$ | $\mathbf{0.0057 \pm 3 \times 10^{-3}}$ | $2.2 \pm 1 \times 10^{-1}$ | $0.016 \pm 9 \times 10^{-3}$ |
| CPA$^*$ (scGPT) | $0.70 \pm 2 \times 10^{-2}$ | $0.025 \pm 6 \times 10^{-3}$ | - | - |
| SAMS-VAE$^*$ | $0.45 \pm 2 \times 10^{-2}$ | $0.021 \pm 5 \times 10^{-3}$ | $1.9 \pm 3 \times 10^{-2}$ | $0.000 \pm 0$ |
| SAMS-VAE$^*$ (S) | $\mathbf{0.78 \pm 6 \times 10^{-3}}$ | $0.019 \pm 5 \times 10^{-3}$ | $\mathbf{0.74 \pm 5 \times 10^{-2}}$ | $0.028 \pm 6 \times 10^{-3}$ |
| Biolord$^*$ | $0.41 \pm 2 \times 10^{-2}$ | $0.027 \pm 1 \times 10^{-3}$ | $1.6 \pm 5 \times 10^{-3}$ | $0.000 \pm 0$ |
| GEARS | $0.44 \pm 5 \times 10^{-3}$ | $0.051 \pm 1 \times 10^{-2}$ | - | - |
| LA | $\mathbf{0.79 \pm 1 \times 10^{-2}}$ | $\mathbf{0.005 \pm 2 \times 10^{-3}}$ | $3.2 \pm 6 \times 10^{-3}$ | $0.000 \pm 3 \times 10^{-4}$ |
| LA (scGPT) | $0.77 \pm 4 \times 10^{-3}$ | $0.0085 \pm 1 \times 10^{-3}$ | - | - |
| Decoder | $0.73 \pm 2 \times 10^{-2}$ | $0.017 \pm 6 \times 10^{-3}$ | $3.2 \pm 4 \times 10^{-3}$ | - |
| Linear | $0.60 \pm 2 \times 10^{-2}$ | $0.035 \pm 4 \times 10^{-3}$ | $1.2 \pm 4 \times 10^{-2}$ | $0.018 \pm 2 \times 10^{-3}$ |

With the `OP3` dataset, we assess models' ability to transfer small molecule effects across heterogeneous PBMC cell types, again finding similar trends in model performance. The gap between CPA/SAMS-VAE and the Latent Additive/Decoder models' performance on the MMD PCA metric was larger, suggesting that the `OP3` dataset contains greater within-cell-type heterogeneity compared to the `Srivatsan20` or `Frangieh21` datasets. Additional details can be found in Appendix C.4

Our simpler baselines, in particular the Latent Additive (LA) model, offer competitive performance at a fraction of training cost. These baseline models also scale well with larger datasets that contain more complex covariates, as shown in the Section 5.3 with the `McFalineFigueroa23` and `Jiang24` datasets, where they outperform the more complex VAEs. However, they underperform more complex models when evaluated on the MMD metrics, suggesting they only learn an average response to perturbation. Thus, it is important to evaluate models across a diverse set of datasets and metrics, which we offer in PerturBench.

## 5.2 Predicting Combinatorial Gene Over-expression Effects

Next we discuss model performance on the *combo prediction* task using the `Norman19` dataset, which contains both single and dual genetic perturbations. The models train on all single perturbations and 30% of the dual perturbations, with the remaining 70% of duals held out for validation and testing. We summarize the results in Table 3.

The Latent Additive, Decoder-Only, and a subset of published models outperform the linear model across all metrics, suggesting that deep learning approaches can capture some non-linear interactions in the `Norman19` dataset. However, the linear model performance performance is still relatively strong, suggesting most dual perturbation effects are linear.

Again, removing the adversary component or sparsity constraint leads to better performance, as is the case of CPA$^*$ (noAdv) and SAMS-VAE$^*$ (S). Both models perform largely on par with our best baseline Latent Additive model. Overall, `Norman19` offers clean perturbation effects, ideal for sanity checking models during development.

The autoencoder architectures again outperform the baseline models in the MMD PCA metric. However, the gap is smaller than the gap observed in the `Srivatsan20`, `Frangieh21`, or `OP3` datsets, suggesting less within-cell-line heterogeneity in the `Norman19` dataset.

## 5.3 Effect of Data Scaling

In this section, we report the *data scaling* results, assessing whether models can take advantage of additional training data to better generalize perturbation effects across biological states. We use the `McFalineFigueroa23` dataset that contains 3 cell lines with 5 chemical perturbations and 525 genetic perturbations. We treat unique cell line and chemical perturbation as a separate biological

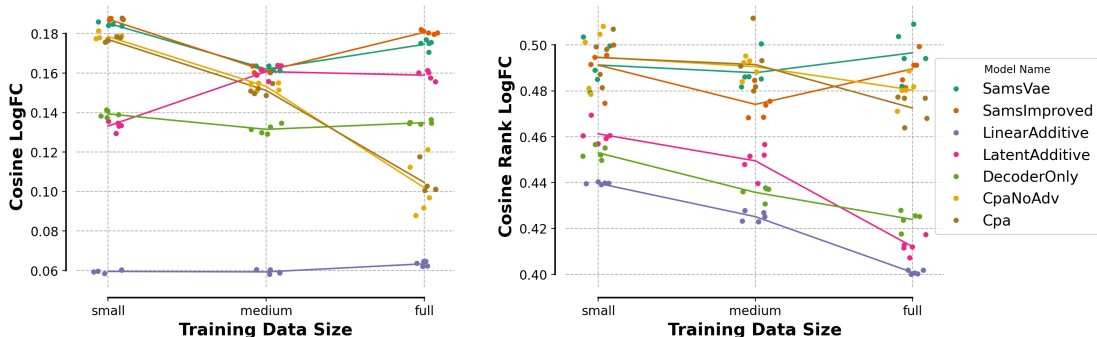

Figure 3: Scaling of cosine similarity (left) and its rank (right) with increasing size of data included in the training process ($x$-axis) for several perturbation response models. Points represent results on test data for 5 different seeds, the line represent their average.

Table 4: Results of the `Jiang24` experiment which contains 30 different biological states (6 cell lines & 5 cytokine treatments), with 70% of perturbations from 9 states held out for validation/testing. Results are reported as mean $\pm$ one standard deviation. Best performance per metric is indicated in bold.

| Model | Cosine logFC | Cosine LogFC rank | MMD PCA | DEG recall |
|---|---|---|---|---|
| CPA* | $0.60 \pm 7 \times 10^{-4}$ | $0.40 \pm 9 \times 10^{-3}$ | $2.5 \pm 4 \times 10^{-3}$ | $0.004 \pm 2 \times 10^{-3}$ |
| CPA* (noAdv) | $0.60 \pm 2 \times 10^{-3}$ | $0.39 \pm 1 \times 10^{-2}$ | $2.5 \pm 3 \times 10^{-3}$ | $0.005 \pm 6 \times 10^{-4}$ |
| SAMS-VAE* | $0.59 \pm 3 \times 10^{-3}$ | $0.45 \pm 9 \times 10^{-3}$ | $\mathbf{0.32 \pm 5 \times 10^{-3}}$ | $0.000 \pm 4 \times 10^{-5}$ |
| SAMS-VAE* (S) | $0.57 \pm 5 \times 10^{-2}$ | $0.42 \pm 2 \times 10^{-2}$ | $1.6 \pm 1 \times 10^{0}$ | $0.002 \pm 1 \times 10^{-4}$ |
| LA | $0.47 \pm 1 \times 10^{-3}$ | $0.38 \pm 6 \times 10^{-3}$ | $2.6 \pm 2 \times 10^{-3}$ | $0.001 \pm 5 \times 10^{-4}$ |
| Decoder | $\mathbf{0.64 \pm 8 \times 10^{-4}}$ | $\mathbf{0.32 \pm 8 \times 10^{-3}}$ | $2.6 \pm 2 \times 10^{-3}$ | - |
| Linear | $0.17 \pm 8 \times 10^{-5}$ | $0.34 \pm 2 \times 10^{-3}$ | $1.3 \pm 2 \times 10^{-3}$ | $0.003 \pm 2 \times 10^{-4}$ |

state, resulting in 15 total states. To test whether adding biological states improves performance, we construct nested subsets of the dataset ($small \subset medium \subset full$), all sharing the same validation and test sets. Each of the subsets contains more biological states (details in Appendix D.4).

We find all models tend to improve with more training data in both cosine LogFC and its rank metric (see Figure 3), with the exception of the CPA* and SAMS-VAE* models. As the training data increases, CPA* and CPA* (noAdv) perform worse on the cosine LogFC metric, while SAMS-VAE* and SAMS-VAE* (S) perform worse on the rank metric. Our baseline models, especially the simple Latent Additive model, generally outperform the more complex CPA* and SAMS-VAE* models, especially on the rank metric and with larger training data sizes. This suggests that the simpler baselines may scale to large datasets better than the more complex published models.

We further assessed how these models perform on large datasets with complex covariates by applying our benchmarking suite to the `Jiang24` dataset, a large, 1.6 million cell dataset with complex covariates. The dataset contained 6 cell lines with 5 unique cytokine treatments, which we modeled as 30 distinct biological states and 219 genetic perturbations. However, the set of perturbations applied was different for each cytokine treatment. We used a similar splitting strategy where we held out 70% of the covariates from 9 cell states for validation/testing.

The results are summarized in Table 4 where the Decoder-Only model performs the best on average response and SAMS-VAE* performs best on the MMD PCA metric. Here, none of the published models perform on-par with our simple baselines, particularly on the rank metrics which indicates these more sophisticated models are unable to generate perturbation-specific responses. There is less of a gap between the more complex autoencoder architectures and the baseline models in the MMD PCA metric, suggesting less within-covariate heterogeneity compared to other datasets. These results

align with our earlier observation that simpler architectures with fewer modeling assumptions benefit more from additional training data.

# 6 Discussion

**Summary and Limitations**    Our study shows that for predicting average perturbation effects, there is no model, or even class of models, that is clearly better than others. In smaller datasets, methods such as CPA* (noAdv) and SAMS-VAE* (S) exhibit the best performance. However, these methods do not scale well in larger datasets, where they are outperformed by our Latent Additive or Decoder-Only models using the *matching* approach. The variance in performance across datasets highlights the need to develop more versatile and universally robust models and we hope that our perturbation framework will be helpful for developing and benchmarking these future models.

In terms of predicting the distribution of perturbation responses, as measured by the MMD metrics, autoencoder models such as CPA* and SAMS-VAE* outperform the Latent Additive and Decoder-Only baselines. The gap in MMD performance depends on the dataset, suggesting that some datasets have less heterogeneity within biological states. More recent model architectures such as STATE and CellFlow may be able to predict the full distribution of perturbation responses while also maintaining strong performance on average effect metrics [2, 28]

We also find that ablating the adversarial component in CPA* and the sparsity-inducing component in SAMS-VAE* improve performance over the original models. Since it is highly probable that assumptions made and tested in one dataset may not hold true in another, comprehensive ablation experiments across diverse datasets and metrics as are provided in Perturbench will be a essential to developing more versatile and robust model architectures.

Our simple Decoder-Only models with no gene expression input generally perform well, outperforming all other models on the Jiang24 dataset. This potentially points to new architectures that emphasize the perturbation encoder, assuming the information present in the gene expression input may not be as rich as the perturbation and covariate information. Other possible explanations include inadequate control matching due to heterogeneity in the control cells not captured by covariate labels, as well as noise in the single-cell RNA readouts or the perturbation labels. Specifically, some cells with a perturbation label may not have a robust target gene knockdown in CRISPR experiments [32].

Models perform worse on the larger datasets and tasks, specifically McFalineFigueroa23 (*data scaling*) and Jiang24. For example, on the *data scaling* task, no model achieved a rank metric below $0.4$ (where $0.5$ is random predictor). This could be due to both the intrinsic difficulty of the task, and the amount of measurement or biological noise present in a dataset. For the *data scaling* task, most models performed better with more data, potentially suggesting perturbation models follow scaling laws [25].

Given the diversity of implementations among public models, we aimed to assess the core components of each model. Thus, our benchmarking results should be interpreted as an assessment of how these core components perform rather than a perfectly accurate recreation of the public model implementation (see Appendix D.5 for details).

**Benchmarking codebase**    We provide three main components: datasets and dataloaders, a model development framework, and an evaluation API with metrics (Appendix A). Each component can be used together or individually. For example, a user who just wants to benchmark predictions generated by an existing model can use the evaluation API. Whereas a user who wants to develop a new model with our model framework can use the entire codebase. Each component is extensible, making it easy for users to add new datasets, models, and metrics. The codebase can be found on GitHub at https://github.com/altoslabs/perturbench/.

**Conclusion**    The perturbation response modeling field holds great promise in searching the truly vast space of genetic and chemical perturbations to find disease targets and potential therapeutics. In this work, we bring together state-of-the-art models in a unified framework with thorough evaluation. We benchmarked individual model components through ablation studies, assessed performance on different datasets at different scales, and developed rank metrics that capture key downstream applications and model failure modes. Finally, we hope that our modular codebase will prove valuable in future model development and benchmarking efforts.

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

# Appendix

## A   Software Framework: The PerturBench Library

The PerturBench library is designed to encapsulate the essential elements of perturbation modeling, prioritizing reusability and flexibility for researchers. It integrates seamlessly with leading Python-based machine learning frameworks including PyTorch, PyTorch Lightning, and Hydra, as well as cutting-edge single-cell analysis libraries such as Scanpy and AnnData. Our design choices streamline the training of innovative model architectures and the assessment of both existing and novel techniques across a comprehensive range of benchmarks and datasets. The library is structured into three core modules: `data`, `model`, and `analysis`. These modules are engineered to work together to support a variety of applications, from complete model development to modular use for integration with other tools and analytical assessments. Subsequent sections will detail the primary abstractions each module offers, illustrating their practicality and adaptability for diverse research tasks.

### A.1   Foundational Frameworks

PerturBench leverages contemporary machine learning and single-cell analysis libraries that are prevalent within their respective research communities. This strategic choice is intended to lower the adoption barrier for the proposed benchmark. Additionally, these libraries offer comprehensive guidelines on usage patterns and best practices, which serve to inform the organizational structure of the code.

**Pytorch:**   is one the most widely spread neural network libraries [39]. Its core functionality is to build computational graphs with support for efficient auto differentiation. Using this autodiff engine, Pytorch then provides abstractions to build and optimize various neural networks and ML algorithms. In addition, it provides utilities to load and serve data under different training regimes. These concepts are captured within the `torch.utils.data.Dataset` and `torch.utils.data.DataLoader` abstractions. We use these to implement our `perturbench.data` module.

**Pytorch Lightning:**   While Pytorch provides most of the functionality to train any neural network model, it could still be a challenging task to write training and evaluation code that can be portable across different platforms, have minimal boiler plate code, and be easy to read and understand. Pytorch Lighting is a library that builds on top Pytorch Lighting to provide [17]: 1) hardware agnostic model implementations, 2) clear easy to read codebase with minimal boiler plate code, 3) reproducible experiments, and 4) integration with popular machine learning tools. We wrap our models and data into Lightning's `LightningModule` and `LightningDataModule` to abstract away most of the code for managing model training and serving data. Then we leverage Lightning's `Trainer` that abstracts the various traning loops to write generic train and evaluation scripts. Furthermore, Lightning's `Callback` to integrate various logging libraries such as TensorBoard.

**Hydra:**   A complex benchmarking suite needs to configure its large number of components and to provide a simple summary for reproducing any experiment. In `pertubench`, we utilize Hydra [59] for managing configuration files. Hydra provides a hierarchical configuration system that can be composed based on the components of the system being configured. In addition, it provides convenient tools such as a command line interface (cli) with auto-completion, support for HPO via optuna, and basic type checking.

**AnnData**   We use AnnData as our primary format for storing and interacting with single cell RNA-seq datasets [57]. Each AnnData object contains a single cell gene expression matrix with associated cell level metadata such as perturbation and covariates, as well as gene level metadata such as gene name and ID. Our data module expects single cell datasets stored as AnnData h5ad files and our analysis module expects model predictions in the form of AnnData objects.

## A.2 Data Abstractions

The library is built around the `Example` abstraction given in Listing 1 that represents a single datum and its batched version. This data structure contains the necessary fields required for the training and evaluation of perturbation prediction models and serves to unify the model/data API. Each example has two required fields: a `gene_expression` 1D tensor that contains the gene expression levels, and the `perturbations` that has the list perturbation names that has been applied to this cell. In addition, the example contains some optional fields that support more complex functionality like using pre-computed embeddings in the `extra` field, or control matching via `controls`. An ordered list of gene names can be provided in `gene_names` such that it is ordered according to the provided gene counts in `gene_expression`.

```python
class Example(NamedTuple):
    """Single Cell Expression Example."""

    gene_expression: Tensor
    perturbations: list[str]
    covariates: dict[str, str] | None = None
    controls: Tensor | None = None
    gene_names: list[str]
    extra: dict[str, Any]
```

Listing 1: Data structure representing a single example.

For training, we provide two types of pytorch datasets. The `SingleCellPerturbation` class represents a single cell RNA-seq dataset (Listing 2) and the `SingleCellPerturbationWithControls` class adds control matching functionality, sampling a matched control cell for every perturbed cell (Listing 3).

```python
class SingleCellPerturbation(Dataset):
    """Single Cell Perturbation Dataset."""

    gene_expression: Tensor
    perturbations: list[list[str]]
    covariates: dict[str, list[str]] | None = None
    cell_ids: list[str] | None = None
    gene_names: list[str] | None = None
    transforms: Callable | None = None
    extra: dict[str, Any]

    # factory method
    @staticmethod
    def from_anndata(
        adata: ad.AnnData,
        perturbation_key: str,
        perturbation_combination_delimiter: str | None,
        covariate_keys: list[str] | None = None,
        perturbation_control_value: str | None = None,
        embedding_key: str | None = None,
    ) -> tuple[SingleCellPerturbation, dict[str, Any]]:
        ...
```

Listing 2: Pytorch dataset classes for training.

```
1  class SingleCellPerturbationWithControls(SingleCellPerturbation):
2      """Single Cell Perturbation Dataset with matched controls."""
3
4      control_ids: Sequence[str] | None = None
5      control_indexes: Map(CovariateDict, list[int])
6      control_expression: Tensor
7
8      # factory method
9      @staticmethod
10     def from_anndata(
11         adata: ad.AnnData,
12         perturbation_key: str,
13         perturbation_combination_delimiter: str | None,
14         covariate_keys: list[str] | None = None,
15         perturbation_control_value: str | None = None,
16         embedding_key: str | None = None,
17     ) -> tuple[SingleCellPerturbation, dict[str, Any]]:
18         ...
```

Listing 3: Pytorch dataset classes for training.

For inference, we provide two additional types of pytorch datasets. The `Counterfactual` dataset represents a desired set of counterfactual predictions. Since these counterfactual predictions are applied to unperturbed control cells, we only need to store control cell expression values. A single item of this dataset is a counterfactual perturbation applied to set of control cells with will return a `Batch` of data with the control cell expression, control covariates, and desired perturbation.

To evaluate counterfactual predictions, we also provide the `CounterfactualWithReference` class which inherits from the `Counterfactual` class. In additional to providing a `Batch` of control cells with covariates and the desired perturbation, this class also provides an AnnData object with the gene expression values for the perturbed cells corresponding to the covariates and desired perturbations. This enables us to use our suite of benchmarking metrics to compare the model predictions with the observed data. We provide the classes in Listing 4.

```
1  class Counterfactual(Dataset):
2      """Counterfactual Dataset."""
3      # Desired counterfactual perturbations
4      perturbations: Sequence[list[str]]
5      covariates: dict[str, Sequence[str]]
6      control_expression: SparseMatrix
7      control_indexes: FrozenDictKeyMap
8      gene_names: Sequence[str] | None = None
9      transforms: InitVar[Callable | Sequence[Callable] | None] = field(
       default=None)
10     info: dict[str, Any] | None = None
11     control_embeddings: np.ndarray | None = None
12
13 class CounterfactualWithReference(Counterfactual):
14     """Counterfactual Dataset with matched Reference Data."""
15     # A map from a unique perturbation and set of covariates to
       indexes
16     # in the reference_adata (i.e. all indexes that contain k562 cells
17     # with AGR2 knocked down)
18     reference_indexes: dict[str, FrozenDictKeyMap] | None = None
19     # An AnnData object containing the observed perturbational dataset
20     # matching the desired counterfactual predictions
21     reference_adata: ad.AnnData | None = None
```

Listing 4: Pytorch dataset classes for inference.

## A.3 Data Splitting

We implement a datasplitter class that can generate three types of datasplits:

1. Cross covariate splits that ask a model to predict a perturbation's effect in covariate(s) that were not in the training split. The model will have seen other perturbation in the covariate(s).

2. Combinatorial splits that ask a model to predict the effect of multiple perturbations. The model will have seen the individual perturbations and some other combinations.

3. Inverse combinatorial splits that ask a model to predict the effect of a single perturbation when it has seen a dual perturbation and the other single perturbation.

We design a data splitter with two parameters that allow us to curate the splits: (1) The maximum number, $m$, of cell types (covariates) to hold out. We randomly hold out between one and $m$ cell types (sampled uniformly). The more cell types held out, the more challenging the task becomes due to fewer training cell types. (2) The total fraction of perturbations held out per cell type, $f$. A larger fraction makes it more difficult for the model to generate accurate predictions. The datasplitter can also read in custom splits from disk as a csv file.

## A.4 Model Abstraction: Model Base Class

We implement a base model class, `PerturbationModel`, that abstracts away common model components that we describe in Listing 5. This class specifically contains:

- A default optimizer
- A training record that contains the data transforms and other key metadata needed for training and inference
- Methods for generating and evaluating counterfactual predictions

```python
class PerturbationModel(L.LightningModule, ABC):
    """A base model class for perturbation prediction models."""
    training_record: dict = {
        'transform': None,
        'train_context': None,
        'n_total_covs': None,
    }
    evaluation_config: DictConfig | None = None
    summary_metrics: pd.DataFrame | None = None
    prediction_output_path: str | None = None

    def configure_optimizers(self):
        """Base optimizer for lightning Trainer."""

    def predict_step(
        self,
        data_tuple: tuple[Batch, pd.DataFrame],
        batch_idx: int,
    ) -> ad.AnnData | None:
        """Given a batch of data, predict the counterfactual perturbed
    """

    def test_step(
        self,
        data_tuple: tuple[Batch, pd.DataFrame, ad.AnnData],
        batch_idx: int,
    ):
        """Run evaluation on a Batch of counterfactual predictions and
        matched observed predictions."""

    def on_test_end(self) -> None:
        """Run rank evaluations (if specified) and summarize
    benchmarking
```

```
32        metrics."""
33
34    @abstractmethod
35    def predict(self, counterfactual_batch: Batch) -> torch.Tensor:
36        """Given a counterfactual_batch of data,
37        predict the counterfactual perturbed expression.
38        """
```

Listing 5: Pytorch dataset classes for inference.

## A.5   Evaluation

All models that inherit from the base `PerturbationModel` class will be able to run evaluation using the Pytorch Lightning trainer test step. These evaluations can be configured via Hydra if using our `train.py` script and evaluations can be run automatically after training completes. For users who only want to use our evaluation metrics, we offer a kaggle style evaluation API that takes as input model predictions as an AnnData object, with an example in Listing 6.

```
1  from perturbench.analysis.benchmarks.evaluator import Evaluator
2
3  # List available tasks
4  print(Evaluator.list_tasks())
5
6  # Select an evaluation task
7  evaluator = Evaluator(
8      task = "sciplex3-transfer",
9  )
10 # The input format of the Evaluator class is a
11 # dictionary of model predictions stored as AnnData objects
12 input_dict = {"CPA_pred": cpa_pred} # cpa_pred is an AnnData object
13 result_df = evaluator.evaluate(input_dict)
14 print(result_df) # Summary dataframe with evaluation metrics
```

Listing 6: Evaluation API usage example.

# B Additional Modeling Background

## B.1 Perturbation embeddings

**Drug Embeddings** It can be beneficial to use pre-trained embeddings to enable or enhance predictive performance of perturbation models, for instance, ESM embeddings for gene expression [45]. The performance of these models in predicting unseen perturbations is dependent on the quality of the perturbation representation, which is itself a complex task [21, 48, 45] and outside the scope of this study. GEARS uses gene co-expression to build a gene to gene graph [46], PerturbNet uses a perturbation encoder network to encode perturbations into a lower dimensional embedding [60]. For drug perturbations, PerturbNet uses a structure encoder and for genetic perturbations, it models the gene as a multi-hot vector over all gene ontology annotations. The authors of CPA include a variation to their original model that embeds drugs into a lower dimensional space using molecular features [34]. scFoundation leverages GEARS but instead of constructing the graph using static gene coexpression, it uses the gene embeddings for a given cell to create a gene-gene graph [20]. The performance of these models in predicting unseen perturbations is dependent on the quality of the perturbation representation, which is itself a complex task [21, 48, 45] and outside the scope of this study.

## C  Further Results

### C.1  Additional Srivatsan20 metrics

Table 5: Additional `Srivatsan20` metrics

| Model | RMSE mean | RMSE mean rank | MMD |
|---|---|---|---|
| CPA* | $0.020 \pm 3 \times 10^{-4}$ | $0.16 \pm 7 \times 10^{-3}$ | $2.4 \pm 1 \times 10^{-2}$ |
| CPA* (noAdv) | $0.020 \pm 1 \times 10^{-4}$ | $\mathbf{0.10 \pm 5 \times 10^{-3}}$ | $2.3 \pm 3 \times 10^{-2}$ |
| CPA* (scGPT) | $0.020 \pm 3 \times 10^{-4}$ | $0.16 \pm 2 \times 10^{-2}$ | - |
| SAMS-VAE* | $0.023 \pm 8 \times 10^{-5}$ | $0.17 \pm 1 \times 10^{-2}$ | $2.5 \pm 2 \times 10^{-2}$ |
| SAMS-VAE* (S) | $\mathbf{0.018 \pm 3 \times 10^{-4}}$ | $0.13 \pm 1 \times 10^{-2}$ | $2.9 \pm 1 \times 10^{-2}$ |
| Biolord* | $0.086 \pm 4 \times 10^{-2}$ | $0.35 \pm 1 \times 10^{-1}$ | $4.9 \pm 3 \times 10^{0}$ |
| LA | $\mathbf{0.018 \pm 6 \times 10^{-5}}$ | $0.15 \pm 3 \times 10^{-3}$ | $4.3 \pm 2 \times 10^{-1}$ |
| LA (scGPT) | $\mathbf{0.017 \pm 1 \times 10^{-4}}$ | $0.14 \pm 5 \times 10^{-3}$ | - |
| Decoder | $\mathbf{0.018 \pm 1 \times 10^{-4}}$ | $0.14 \pm 7 \times 10^{-3}$ | $4.2 \pm 4 \times 10^{-3}$ |
| Decoder (Cov) | $0.023 \pm 3 \times 10^{-5}$ | $0.50 \pm 4 \times 10^{-2}$ | - |
| Linear | $0.030 \pm 5 \times 10^{-4}$ | $0.27 \pm 2 \times 10^{-3}$ | $\mathbf{2.2 \pm 7 \times 10^{-3}}$ |

### C.2  Additional Norman19 metrics

Table 6: Additional `Norman19` metrics

| Model | RMSE mean | RMSE mean rank | MMD |
|---|---|---|---|
| CPA* | $0.046 \pm 5 \times 10^{-4}$ | $0.019 \pm 3 \times 10^{-3}$ | $5.6 \pm 3 \times 10^{-2}$ |
| CPA* (noAdv) | $0.046 \pm 1 \times 10^{-3}$ | $0.017 \pm 3 \times 10^{-3}$ | $5.5 \pm 1 \times 10^{-1}$ |
| CPA* (scGPT) | $0.053 \pm 1 \times 10^{-3}$ | $0.036 \pm 3 \times 10^{-3}$ | - |
| SAMS-VAE* | $0.084 \pm 7 \times 10^{-4}$ | $0.026 \pm 2 \times 10^{-3}$ | $4.1 \pm 4 \times 10^{-2}$ |
| SAMS-VAE* (S) | $0.047 \pm 2 \times 10^{-3}$ | $0.030 \pm 7 \times 10^{-3}$ | $3.3 \pm 5 \times 10^{-2}$ |
| Biolord* | $0.086 \pm 6 \times 10^{-4}$ | $0.028 \pm 1 \times 10^{-3}$ | $2.8 \pm 5 \times 10^{-3}$ |
| GEARS | $0.069 \pm 1 \times 10^{-3}$ | $0.055 \pm 6 \times 10^{-3}$ | - |
| LA | $\mathbf{0.043 \pm 4 \times 10^{-4}}$ | $\mathbf{0.014 \pm 1 \times 10^{-3}}$ | $6.7 \pm 4 \times 10^{-3}$ |
| LA (scGPT) | $\mathbf{0.044 \pm 4 \times 10^{-4}}$ | $\mathbf{0.013 \pm 2 \times 10^{-3}}$ | - |
| Decoder | $\mathbf{0.043 \pm 3 \times 10^{-4}}$ | $\mathbf{0.014 \pm 4 \times 10^{-4}}$ | $6.7 \pm 6 \times 10^{-3}$ |
| Linear | $0.057 \pm 3 \times 10^{-3}$ | $0.016 \pm 8 \times 10^{-4}$ | $\mathbf{2.5 \pm 4 \times 10^{-2}}$ |

### C.3  Generalizing from less complex to more complex biological systems

We also apply PerturBench to a highly relevant real-world task: predicting perturbation effects in more complex disease system using effects measured in less complex systems. The `Frangieh21` dataset contains 3 biological systems: primary melanoma cells cultured alone, or with IFN$\gamma$, or co-cultured with tumor infiltrating immune cells. We held out 70% of the perturbations in the co-culture system and used the remaining perturbations as well as all perturbations in the other systems as training.

The results are summarized in Table 7. Our baseline models such as LA and Decoder generally perform better on the rank metrics, whereas more sophisticated models such as SAMS-VAE* (S) perform better on cosine LogFC and RMSE. Again, model simplification consistently results in performance gains, as in the case of CPA* (noAdv) and SAMS-VAE* (S). Overall, `Frangieh21` shows that biological heterogeneity of datasets has a major impact on model comparison, again highlighting the need to include diverse datasets for benchmark.

Aside from this note, we also observe that the vanilla SAMS-VAE* suffers from mode collapse, as indicated by its near-random rank scores on cosine LogFC and RMSE. Indeed, we notice the model is generating generic expression vectors irrespective of the target perturbation, see details in Appendix C.7. Further dissecting SAMS-VAE* performance, we find that model has learnt near-identical embedding vectors for any perturbations. This suggests that despite the theoretical advantage of sparse additive mechanism, effectively optimizing the model in practice remains a non-trivial question, and can lead to degenerate scenarios which fortunately can be uncovered by our rank metric.

Table 7: Results of a *covariate transfer* experiment generalizing from less complex biological systems to a more complex co-culture system in the `Frangieh21` dataset.

| Model | Cosine logFC | Cosine LogFC rank | MMD PCA | DEG recall |
|---|---|---|---|---|
| CPA* | $0.10 \pm 7 \times 10^{-3}$ | $0.30 \pm 3 \times 10^{-2}$ | $0.26 \pm 5 \times 10^{-3}$ | $0.000 \pm 0$ |
| CPA* (noAdv) | $0.10 \pm 7 \times 10^{-3}$ | $\mathbf{0.20 \pm 3 \times 10^{-3}}$ | $0.21 \pm 2 \times 10^{-3}$ | $0.000 \pm 0$ |
| CPA* (scGPT) | $0.07 \pm 4 \times 10^{-3}$ | $0.26 \pm 5 \times 10^{-3}$ | - | - |
| SAMS-VAE* | $0.15 \pm 2 \times 10^{-2}$ | $0.48 \pm 2 \times 10^{-2}$ | $0.29 \pm 2 \times 10^{-3}$ | $0.000 \pm 0$ |
| SAMS-VAE* (S) | $\mathbf{0.22 \pm 3 \times 10^{-3}}$ | $0.22 \pm 8 \times 10^{-3}$ | $0.32 \pm 4 \times 10^{-3}$ | $0.000 \pm 0$ |
| BioLord* | $0.12 \pm 9 \times 10^{-3}$ | $0.29 \pm 3 \times 10^{-2}$ | $\mathbf{0.20 \pm 3 \times 10^{-3}}$ | $0.000 \pm 1 \times 10^{-4}$ |
| LA | $0.17 \pm 6 \times 10^{-3}$ | $0.26 \pm 1 \times 10^{-2}$ | $3.1 \pm 3 \times 10^{-2}$ | $0.000 \pm 0$ |
| LA (scGPT) | $0.18 \pm 6 \times 10^{-3}$ | $0.27 \pm 1 \times 10^{-2}$ | - | - |
| Decoder | $0.10 \pm 2 \times 10^{-3}$ | $\mathbf{0.21 \pm 5 \times 10^{-3}}$ | $3.3 \pm 4 \times 10^{-4}$ | - |
| Linear | $0.01 \pm 4 \times 10^{-4}$ | $0.24 \pm 9 \times 10^{-4}$ | $0.79 \pm 6 \times 10^{-3}$ | $0.000 \pm 0$ |

Table 8: Additional `Frangieh21` metrics.

| Model | RMSE mean | RMSE mean rank | MMD |
|---|---|---|---|
| CPA* | $0.027 \pm 1 \times 10^{-4}$ | $0.28 \pm 1 \times 10^{-2}$ | $2.1 \pm 2 \times 10^{-2}$ |
| CPA* (noAdv) | $0.027 \pm 9 \times 10^{-5}$ | $0.19 \pm 3 \times 10^{-2}$ | $1.8 \pm 1 \times 10^{-2}$ |
| CPA* (scGPT) | $0.029 \pm 2 \times 10^{-4}$ | $0.26 \pm 1 \times 10^{-2}$ | - |
| SAMS-VAE* | $0.026 \pm 2 \times 10^{-4}$ | $0.46 \pm 2 \times 10^{-2}$ | $2.2 \pm 1 \times 10^{-2}$ |
| SAMS-VAE* (S) | $\mathbf{0.025 \pm 5 \times 10^{-5}}$ | $0.22 \pm 1 \times 10^{-2}$ | $2.4 \pm 1 \times 10^{-2}$ |
| BioLord* | $0.027 \pm 2 \times 10^{-4}$ | $0.21 \pm 4 \times 10^{-2}$ | $1.1 \pm 2 \times 10^{-2}$ |
| LA | $\mathbf{0.024 \pm 4 \times 10^{-4}}$ | $0.21 \pm 2 \times 10^{-2}$ | $6.4 \pm 3 \times 10^{-2}$ |
| LA (scGPT) | $\mathbf{0.024 \pm 6 \times 10^{-5}}$ | $0.24 \pm 1 \times 10^{-2}$ | - |
| Decoder | $\mathbf{0.025 \pm 4 \times 10^{-5}}$ | $\mathbf{0.15 \pm 4 \times 10^{-4}}$ | $\mathbf{6.6 \pm 8 \times 10^{-4}}$ |
| Linear | $0.043 \pm 7 \times 10^{-5}$ | $0.30 \pm 2 \times 10^{-3}$ | $2.1 \pm 8 \times 10^{-3}$ |

### C.4 Generalizing across cell types in primary tissue dataset

The *covariate transfer* task in the `OP3` dataset targets primary tissue which are Peripheral Blood Mononuclear Cells (PBMCs) collected from three donors. It was originally used in the NeurIPS 2023 Perturbation Prediction Challenge [53]. The purpose of this task is to assess a model's ability to predict the effects of chemical perturbations in held-out B and Myeloid cell types, while training data are primarily composed of perturbation profiles in T and NK cells.

Our results are reported in Table 9 and 10. In general, we see that both implementations of CPA and SAMS-VAE perform competitively — often matching or even surpassing the baseline methods: Latent, Decoder, and Linear models, on both the pseudobulk and distributional metrics in the OP3 dataset task. Ablating the adversarial component of CPA (CPA noAdv) reduces performance on the cosine logFC and RMSE metrics, while slightly improving performance on the rank metrics. Our improved SAMS-VAE implementation which involves ablating the sparse mask performs better on all metrics. The Decoder Only model outperforms the Latent Additive baseline on the rank metrics and is close on the Cosine logFC and RMSE metrics, suggesting that the simpler Decoder Only

architecture that maps directly from perturbation to average perturbed cell response is superior for this OP3 dataset when predicting average responses.

All models again perform poorly on the DEG recall task, similar to what we have seen for other datasets. The CPA, SAMS-VAE abd BioLord models are significantly better than baselines on the MMD metrics, especially when computed in PCA space (MMD PCA). As with the other datasets, models generally perform worse in the gene expression space (MMD GEX). The MMD metrics in both gene expression and PCA space are worse for the OP3 dataset compared to our other datasets, suggesting a more heterogeneous perturbation response which confirms the value of the added OP3 dataset.

Table 9: Main results of the `OP3` experiment. Model performance predicting small molecule perturbation effects across primary PBMC cell types in the `OP3` dataset.

| Model | Cosine LogFC | Cosine LogFC rank | MMD PCA | DEG recall |
|---|---|---|---|---|
| CPA* | $0.39 \pm 6 \times 10^{-3}$ | $0.091 \pm 2 \times 10^{-3}$ | $\mathbf{0.71 \pm 1 \times 10^{-2}}$ | $0.002 \pm 2 \times 10^{-4}$ |
| CPA* (noAdv) | $0.32 \pm 6 \times 10^{-3}$ | $0.076 \pm 3 \times 10^{-3}$ | $0.86 \pm 1 \times 10^{-2}$ | $0.000 \pm 0$ |
| SAMS-VAE* | $0.41 \pm 1 \times 10^{-2}$ | $0.11 \pm 2 \times 10^{-2}$ | $0.90 \pm 3 \times 10^{-2}$ | $0.000 \pm 0$ |
| SAMS-VAE* (S) | $\mathbf{0.43 \pm 2 \times 10^{-3}}$ | $0.079 \pm 7 \times 10^{-3}$ | $0.74 \pm 3 \times 10^{-3}$ | $0.000 \pm 0$ |
| Biolord* | $0.28 \pm 3 \times 10^{-3}$ | $\mathbf{0.051 \pm 5 \times 10^{-3}}$ | $0.92 \pm 5 \times 10^{-3}$ | $0.000 \pm 0$ |
| LA | $0.39 \pm 2 \times 10^{-2}$ | $0.12 \pm 3 \times 10^{-2}$ | $4.1 \pm 5 \times 10^{-2}$ | $0.000 \pm 6 \times 10^{-5}$ |
| Decoder | $0.32 \pm 6 \times 10^{-3}$ | $0.070 \pm 5 \times 10^{-3}$ | $4.4 \pm 1 \times 10^{-2}$ | - |
| Linear | $0.011 \pm 9 \times 10^{-5}$ | $0.14 \pm 4 \times 10^{-4}$ | $2.6 \pm 4 \times 10^{-3}$ | $0.000 \pm 0$ |

Table 10: Additional `OP3` metrics

| Model | RMSE mean | RMSE mean rank | MMD |
|---|---|---|---|
| CPA* | $0.032 \pm 2 \times 10^{-4}$ | $0.062 \pm 4 \times 10^{-3}$ | $5.3 \pm 3 \times 10^{-2}$ |
| CPA* (noAdv) | $0.034 \pm 4 \times 10^{-4}$ | $\mathbf{0.051 \pm 4 \times 10^{-3}}$ | $5.7 \pm 1 \times 10^{-2}$ |
| SAMS-VAE* | $0.034 \pm 5 \times 10^{-4}$ | $0.11 \pm 2 \times 10^{-2}$ | $5.6 \pm 4 \times 10^{-2}$ |
| SAMS-VAE* (S) | $\mathbf{0.031 \pm 1 \times 10^{-4}}$ | $\mathbf{0.051 \pm 4 \times 10^{-3}}$ | $5.6 \pm 9 \times 10^{-3}$ |
| Biolord* | $0.038 \pm 8 \times 10^{-5}$ | $\mathbf{0.051 \pm 6 \times 10^{-3}}$ | $4.6 \pm 2 \times 10^{-2}$ |
| LA | $0.036 \pm 2 \times 10^{-3}$ | $0.14 \pm 3 \times 10^{-2}$ | $11.0 \pm 6 \times 10^{-2}$ |
| Decoder | $0.035 \pm 2 \times 10^{-4}$ | $0.068 \pm 6 \times 10^{-3}$ | $11.0 \pm 5 \times 10^{-3}$ |
| Linear | $0.070 \pm 4 \times 10^{-5}$ | $0.24 \pm 9 \times 10^{-4}$ | $\mathbf{4.0 \pm 7 \times 10^{-3}}$ |

## C.5 Additional Jiang24 metrics

Table 11: Additional `Jiang24` metrics

| Model | RMSE mean | RMSE mean rank | MMD |
|---|---|---|---|
| CPA* | $\mathbf{0.015 \pm 5 \times 10^{-5}}$ | $0.42 \pm 1 \times 10^{-2}$ | $8.0 \pm 2 \times 10^{-3}$ |
| CPA* (noAdv) | $\mathbf{0.015 \pm 4 \times 10^{-5}}$ | $0.40 \pm 1 \times 10^{-2}$ | $8.0 \pm 2 \times 10^{-3}$ |
| SAMS-VAE* | $0.017 \pm 2 \times 10^{-4}$ | $0.45 \pm 6 \times 10^{-3}$ | $4.5 \pm 7 \times 10^{-3}$ |
| SAMS-VAE* (S) | $0.016 \pm 2 \times 10^{-3}$ | $0.42 \pm 3 \times 10^{-2}$ | $6.5 \pm 2 \times 10^{0}$ |
| LA | $\mathbf{0.015 \pm 7 \times 10^{-5}}$ | $0.38 \pm 7 \times 10^{-3}$ | $8.0 \pm 9 \times 10^{-4}$ |
| Decoder | $\mathbf{0.015 \pm 3 \times 10^{-5}}$ | $\mathbf{0.32 \pm 5 \times 10^{-3}}$ | $8.0 \pm 2 \times 10^{-3}$ |
| Linear | $0.038 \pm 5 \times 10^{-5}$ | $0.43 \pm 1 \times 10^{-3}$ | $\mathbf{3.1 \pm 2 \times 10^{-3}}$ |

## C.6 Effect of Data Imbalance

An important consideration with perturbation response models is how robust they are to *imbalanced data* i.e. how evenly data is distributed across covariates. We quantify imbalance using normalized entropy as follows:

$$\text{Imbalance} := 1 - \frac{\sum_{i=1}^{k} \frac{n_i}{n} \log \frac{n_i}{n}}{\log k},$$

where $n_i, i = 1, \ldots, k$ denotes the number of samples in class $i$ and $n = \sum_{i=1}^{n} n_i$ the overall number of observations. The `Srivatsan20` data is perfectly balanced (Imbalance $= 0$), with every perturbation being observed in every cell line. However, *in-silico* machine learning perturbation models often aim to learn generalizable features by using data from multiple sources, which will invariably produce imbalanced datasets.

To test how different models' performance is affected by data imbalance we downsample perturbations per cell line from `Srivatsan20` to construct three sub-datasets with different levels of imbalance (Appendix D.4). The results are summarized in Figure C.1. We observe that when the data is highly balanced, the linear model performs acceptably well, but this does not hold as imbalance increases. Imbalance may therefore be an important criteria for deciding the suitability of a linear model. CPA both with and without scGPT embeddings, is more robust to changes in data balance than the the linear or Latent Additive models. Interestingly, whilst the Latent Additive model is more markedly affected by data imbalance than other models, using scGPT embeddings seems to buffer performance by some extent. The extent to which performance is affected by data imbalance highlights the importance of curated datasets as well as potential mitigation strategies such as oversampling rare cell lines/types [14].

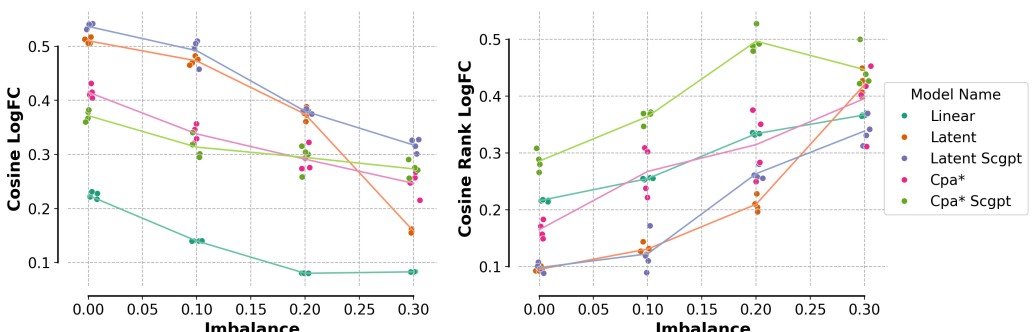

Figure C.1: Cosine similarity of log fold changes (left) and its rank (right) of the models as a function of data balance.

## C.7 Collapse and Rank Metrics

*Mode* or *posterior collapse* is a common failure mode in generative models, notably in Generative Adversarial Networks (GANs) and Variational Autoencoders (VAEs). The problem arises when a generative model inadequately captures the diversity of training data, resulting in repetitive outputs that are effectively collapsing onto a set of limited modes in the data distribution. For VAEs, posterior collapse indicates the latent variables lack meaningful information and are ignored by the decoder. This effectively rids a VAE of any mechanism to control the properties of generated samples, which in practice also leads to mode collapse.

For perturbation response modelling, these issues are particularly concerning as the predicted responses are expected to be perturbation-specific, so that they are able to capture the nuances of perturbation effects across a diverse set of cell types/lines, treatments and other conditions. This is above-all essential for inferring the distinct effect of perturbations on various biological processes, as well as ranking and identifying targets for the disease of interest.

However, as we will demonstrate, traditional measures such as RMSE, cannot distinguish between mode collapse or not, because models that simply predict the the average expression level for a particular cell line can still achieve relatively good performance. This motivates our rank metrics

which can act as a safeguard against such degenerate cases, to some extent. Yet, we reveal its limitations through comparison with the visual assessment, and propose two new metrics that are more diagnostic of mode collapse.

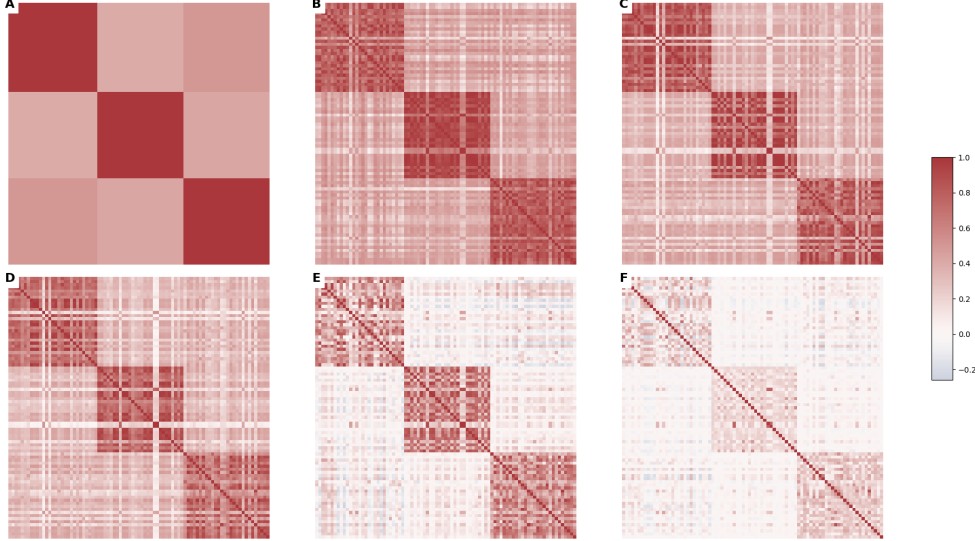

Figure C.2: Cosine similarity matrix based on log-fold changes predicted, between every pair of perturbation-covariate combination in `Srivatsan20` dataset. All models are hyperparameters optimised. **A**) `DecoderOnly` model with only covariates as input. **B**) `DecoderOnly` model with covariates and perturbations as input. **C**) CPA*. **D**) CPA* (noAdv). **E**) SAMS-VAE* (S). **F**) true log-fold changes in the dataset. Diagnoal blocks correspond to cell lines: A549, K562, MCF-7.

Table 12: Numerical quantification of mode collapse for models presented in `Srivatsan20` dataset.

| Model | RMSE mean | RMSE mean rank | RMSE mean transposed-rank | Matrix distance |
|---|---|---|---|---|
| DecoderOnly | 0.026 | 0.488 | 0.482 | 49.906 |
| DecoderOnly (+ perturbations) | 0.021 | 0.116 | 0.232 | 40.645 |
| CPA* | 0.022 | 0.104 | 0.323 | 36.343 |
| CPA* (noAdv) | 0.021 | 0.098 | 0.337 | 32.238 |
| SAMS-VAE* (S) | 0.020 | 0.111 | 0.179 | 19.301 |

### C.7.1 Mode collapse in `Srivatsan20` dataset

Consider an example based on the `Srivatsan20` dataset which measures the perturbational effects of small molecules applied to three cancer cell lines: A549, K562, MCF-7. We look at five models: 1) our `DecoderOnly` with *only* cell line as input, 2) our `DecoderOnly` with cell line *and* perturbations as input, 3) the vanilla CPA*, 4) CPA* with no adversarial components, and 5) the simplified version of SAMS-VAE*. In addition, we compare the model predictions to the experimental data.

Results are shown in Figure C.2. Of the 85 unique cell line and perturbation combinations in the validation split, we measure the similarity between the predicted log-fold changes for every pair of such combinations. In total, these similarities form a $85 \times 85$ matrix which is shown as a heatmap.

The experimental data in **F**) shows the similarity of the log-fold changes between different perturbations to be small within each cell line (three diagonal blocks correspond to the three cell lines in `Srivatsan20` dataset), and even smaller between cell lines. In comparison, the `DecoderOnly` model with only cell line as input (panel **A**) shows its predictions are (unsurprisingly) identical for any perturbation within a cell line, which is a prime example of mode collapse. Qualitatively, from **A**) to **E**), models have shown a general declining trend in mode collapse, with the simplified SAMS-VAE* (S) suffering the least mode collapse.

However, we point our readers to the RMSE metrics reported in Table 12, which are of roughly similar order of magnitude for all models. This means judging by the RMSE alone, it is impossible to tell which model suffers from mode collapse. On the other hand, our rank metric distinguishes the degenerative case from the rest, but it is still not sensitive enough to discern the severity of mode collapse.

### C.7.2  Diagnostic metrics for mode collapse

To come up with metrics more indicative of mode collapse, we propose two other metrics which as shown in Table 12, are more strongly correlated with the visual assessment. The first one is called transposed-rank, defined slightly different than our vanilla rank metric:

$$\text{rank}^T_{\text{average}} := \frac{1}{p} \sum_{i=1}^{p} \text{rank}^T(\hat{x}_i), \quad \text{rank}^T(\hat{x}_i) := \frac{1}{p-1} \sum_{\substack{1 \le j \le p \\ j \ne i}} \mathbb{I}(\text{dist}(\hat{x}_i, x_j) \le \text{dist}(\hat{x}_i, x_i)), \tag{3}$$

where $p$ is the number of perturbations that are being modelled, $\hat{x}_i, x_i$ are the predicted and observed (average) expression value of perturbation $i$, and $\text{dist}$ is a generic distance.

Compared to Eq. 2, the new transposed-rank metric ranks the observed expression on a per prediction basis, whereas in the original rank metric, it is vice-versa. We hypothesize that ranking the observations on a per-prediction basis will make the test more challenging thus exposing the weakness of a mode-collapsed model, because in this case the observed expressions have more variance than the predictions.

Finally, we propose a matrix distance based metric which directly measures the difference between the two similarity matrices:

$$\text{distance}(\hat{S}_{\text{cosine}}, S_{\text{cosine}}) = \|\hat{S}_{\text{cosine}} - S_{\text{cosine}}\|_{\text{Frobenius}} = \sqrt{\sum_{i=1}^{n} \sum_{j=1}^{n} (\hat{s}_{ij} - s_{ij})^2} \tag{4}$$

where $S_{\text{cosine}}$ is from model prediction and $\hat{S}_{\text{cosine}}$ is from the experimental data.

Of all metrics reported in Table 12, matrix distance aligns the best with the visual assessment of the heatmaps, followed by the transposed-rank. We have implemented the transposed-rank in the PerturbBench GitHub repository for users to assess mode collapse with a stricter metric.

Unlike our original rank metric, the transposed-rank metric can be affected by the scaling of the predictions. For example, adding a constant factor to all model predictions will not affect the $RMSE_{rank}$ but will affect the $RMSE_{transposed\_rank}$. Thus, the original rank metric is useful if a user is mainly interested in whether the model predictions are ordered correctly. The transposed-rank metric is useful for users that want a stricter assessment of whether the predictions are similar to the ground truth perturbations, beyond the ordering of the perturbations. An adapted version of the transposed-rank metric is being used as the Perturbation Discrimination Score (Eq. 5) for the 2025 Virtual Cell Competition [47].

$$\text{Perturbation Discrimination Score} = 1 - \text{rank}^T_{\text{average}} \tag{5}$$

In the future, we could also add the matrix distance measure as an additional approach for specifically measuring mode collapse.

### C.7.3  Mode collapse in `Frangieh21` dataset

So far, we have shown that all models suffer mode collapse in the `Srivatsan20` dataset, but to varying extent. Now, we move on to `Frangieh21`, and demonstrate that similar observation still holds and in particular, the values of transposed-rank and matrix distance in diagnosing mode collapse.

Results are shown in Figure C.3 and Table 13. Models included for investigation are: 1) the vanilla SAMS-VAE* , 2) the simplified SAMS-VAE*, 3) the vanilla CPA*, 4) CPA* with no adversarial components, and 5) our `LatentAdditive` model. Once again, we compare the model predictions to the experimental data in `Frangieh21` which has only one cell line and 87 unique perturbations.

Table 13: Numerical quantification of mode collapse for models presented in `Frangieh21` dataset.

| Model | RMSE mean | RMSE mean rank | RMSE mean transposed-rank | Matrix distance |
|---|---|---|---|---|
| SAMS-VAE* | 0.024 | 0.383 | 0.491 | 81.007 |
| SAMS-VAE* (S) | 0.023 | 0.160 | 0.416 | 54.481 |
| CPA* | 0.024 | 0.236 | 0.484 | 80.434 |
| CPA* (noAdv) | 0.024 | 0.156 | 0.480 | 80.166 |
| LA | 0.023 | 0.127 | 0.413 | 65.308 |

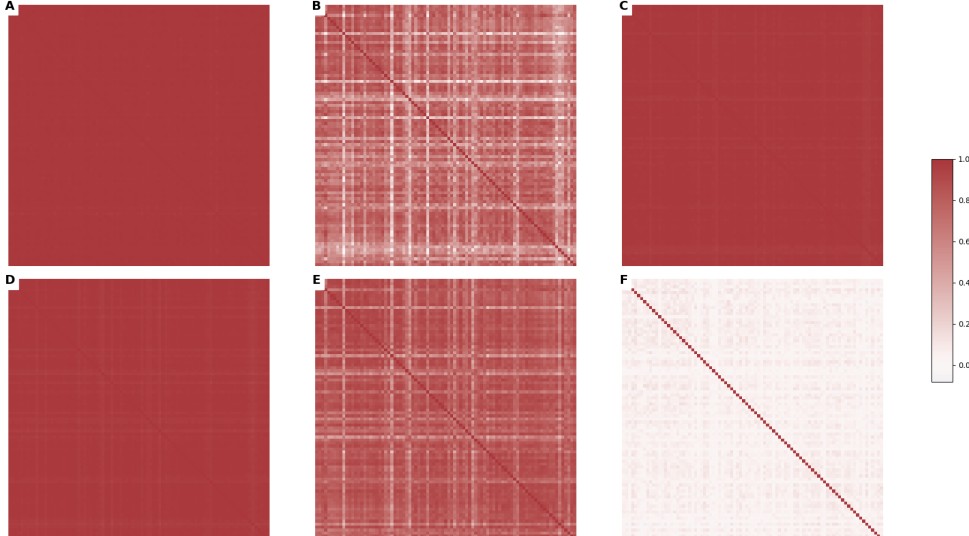

Figure C.3: Cosine similarity matrix based on log-fold changes predicted, between every pair of perturbation in `Frangieh21` dataset. All models are hyperparameters optimised **A**) `SAMS-VAE*`. **B**) `SAMS-VAE* (S)`. **C**) `CPA*`. **D**) `CPA* (noAdv)`. **E**) `LatentAdditive`. **F**) true log-fold changes in the dataset.

The vanilla SAMS-VAE* has indeed mode-collapsed as it is observed in section C.3. CPA* and CPA* (noAdv) also suffer significant mode collapse, despite RMSE rank suggests otherwise. On the other hand, RMSE transposed-rank and matrix distance metrics clearly indicate mode collapse taking place in these models, which aligns with our visual assessment in Figure C.3. This indicates the transposed-rank and matrix distance are also better suited for identifying mode collapse in the `Frangieh21` dataset.

Overall, based on our observations in `Srivatsan20` and `Frangieh21` datasets, we establish that our simple baseline models such as `DecoderOnly` with perturbation and covariate inputs and `LatentAdditive`, as well as the simplified SAMS-VAE* (S), are less prone to mode collapse.

### C.7.4 Verifying CPA implementation using the `Norman19` dataset

To confirm that our implementation of CPA performed at least as well as the public, we verified that our CPA implementation matches or exceeds the published CPA implementation performance on the Norman19 dataset, where CPA (Theis) is the public version of CPA hosted on github and accessed on 07/30/2025, and CPA (PerturBench) is our recreation of CPA. We specifically trained a CPA (Theis) model using their Norman19 example notebook with provided hyperparameters and their version of the Norman19 dataset, using the "split_6" train/val/test split. We then added their version of the Norman19 dataset to PerturBench, and trained our CPA implementation on the same dataset/split. We evaluated both model's performance on the held out "ood" split using PerturBench metrics.

|                    | CPA (Theis) | CPA (PerturBench) |
| ------------------ | ----------- | ----------------- |
| RMSE               | 0.112       | 0.0273            |
| RMSE rank          | 0.427       | 0.00889           |
| Cosine LogFC       | 0.206       | 0.750             |
| Cosine LogFC rank  | 0.262       | 0.0133            |
| DEG Recall         | 0.0987      | 0.384             |
| MMD PCA            | 3.28        | 2.31              |
| MMD                | 4.24        | 3.77              |

We noticed that the choice of loss has a major impact on performance, and models that used a simple MSE loss performed better than Gaussian or Negative Binomial Negative Log Likelihood (NLL) losses. We chose to use the MSE loss in our CPA reproduction, whereas the public model uses NLL losses we believe is the primary reason why our reimplemented CPA performs significantly better than the public CPA implementation. We feel that the use of MSE loss enables us to compare the rest of the model components more on a more even basis.

### C.7.5 Verifying rank metric calibration with a random model

We ran 10 experiments using a null predictor (randomly initialized Latent Additive model) and ran inference using control cells from the Srivatsan20 dataset. When benchmarking against the held out Srivatsan20 perturbations, the null predictor achieved an average RMSE rank metric of 0.4988, confirming that the metric is correctly calibrated.

# D Implementation Details

## D.1 Dataset Summary

**Dataset 1 (Norman19)** *This datasets [38] contains 287 gene overexpression perturbations with 131 containing multiple perturbations in K562 cells. We selected this dataset as it is the largest perturb-seq dataset with combinatorial perturbations so far. This dataset has also been used in several perturbation prediction studies including CPA [34], scGPT [13], SAMS-VAE [7] and Biolord [41].*

**Dataset 2 (Srivatsan20)** *This dataset [51] includes 188 chemical perturbations across the K562, A549, and MCF-7 cell lines. The chemical perturbations were applied at 4 doses but for the purposes of this study, we subset to highest dose only since most of the models we are benchmarking do not have dose response modeling capacity. We selected this dataset to benchmark prediction of chemical perturbations. This dataset has also been used in multiple perturbation prediction studies including CPA [34] and Biolord [41].*

**Dataset 3 (Frangieh21)** *This dataset [18] includes 248 genetic perturbations across 3 melanoma cell conditions that simulate interaction with immune cells. The conditions are: 1) melanoma cells cultured alone, 2) melanoma cells with IFNγ, and 3) melanoma cells co-cultured with tumor infiltrating immune cells. We selected this dataset to benchmark whether a perturbation response prediction model could predict perturbation effects in the more complex co-culture condition using training data from the simpler conditions.*

**Dataset 4 (Jiang24)** *This dataset [23] includes 219 genetic perturbations across 6 cell lines and 5 cytokine treatments (which can be seen as 30 unique biological states). We selected this dataset due to the large number of biological states and the fact that the perturbations were chosen because they had been reported to modulate cytokine signaling. This dataset has not been previously used to benchmark perturbation prediction models.*

**Dataset 5 (McFalineFigueroa23)** *This dataset [37] includes 525 genetic perturbations across 3 cell lines and 5 chemical treatments (which can be seen as 15 unique biological states). We selected this dataset due to the large number of perturbations and the fact that it contains multiple covariates (cell lines and chemical treatments). This dataset has not been previously used to benchmark perturbation prediction models.*

**Dataset 6 (OP3)** *This dataset [53] includes 144 chemical perturbations applied to four annotated primary peripheral blood mononuclear cell (PBMC) types that are T cells, NK cells, B cells and Myeloid cells. The negative control, DMSO (Dimethyl Sulfoxide), is dosed at 14.1 μM. One positive control, Belinostat, is dosed at 0.1 μM. All other perturbations, including the other positive control, Dabrafenib, are dosed at 1.0 μM. This dataset was previously featured in the NeurIPS 2023 perturbation prediction challenge.*

## D.2 Dataset Curation

We download the gene expression counts matrices which are from the original sources of these datasets. Afterwards, we identify the key metadata columns (perturbation, cell line, chemical treatment) and standardize their naming and format across datasets.

## D.3 Dataset Preprocessing

It is a common practice to pre-process perturbation datasets before feeding them into a machine learning pipeline for training or inference. In this section, we describe the data processing that is used by our benchmark.

To ensure we are capturing the most biologically relevant features, we subset the expression vectors to highly variable or differentially expressed genes. Specifically, we keep the top 4000 variable genes using the scanpy `pp.highly_variable_genes` method with `flavor='seurat_v3'`. We also keep the top 25 top differentially expressed genes for every perturbation in every unique set of covariates, using scanpy's `tl.rank_genes_groups` method with default parameters. For datasets with genetic perturbations, we also ensure that the perturbed gene is included in the feature set as well.

For the models that require log-normalization, we apply the default `scanpy` [57] preprocessing pipeline. Specifically, we divide the counts by the total counts in each cell, multiply by a scaling factor of 10,000, and apply a log-transform with a pseudocount of 1, i.e.

$$x_{i,\text{normalized}} = \log\left(1 + \frac{x_i}{\sum_j x_j} \cdot 10^4\right).$$

## D.4 Data Splitting

**McFalineFigueroa23 splits** We manually generate the data scaling splits for the `McFalineFigueroa23` dataset by first selecting 3 covariates where certain perturbations will be held-out. Of the 3 cell lines (a172, t98g, u87mg) and 5 treatments (control, nintedanib, zstk474, lapatinib, trametinib) in `McFalineFigueroa23`, we have specifically selected the following 3 covariates: a172 with nintedanib, t98g with lapatinib, and u87mg with control. Within each of these "held-out covariates", we randomly hold out 70% of perturbations for validation and testing. Some perturbations may be held out across multiple covariates.

To build the small version of the dataset, we select 3 additional covariates that match the cell line and chemical treatment of the "held-out covariates" to add to the training split (a172 with control treatment, t98g with nintedanib, u87mg with lapatinib). To build the medium version of the dataset, we add the remaining 3 covariates that match cell line and chemical treatment to the training split. The large version contains the full dataset — all 15 covariates.

The attached codebase has a python notebook responsible for generating this split: `notebooks/neurips2025/build_data_scaling_splits.ipynb`.

**Jiang24 splits** We hold out 70% of perturbations in all 12 combinations of the following cytokines: IFNG, INS, TGFB and cell lines: k562, mcf7, ht29, hap1. The remaining perturbations are used for training.

The attached codebase has a python notebook responsible for generating this split: `notebooks/neurips2025/build_jiang24_frangieh21_splits.ipynb`.

**Frangieh21 splits** We hold out 70% of the perturbations in the co-culture condition and use the remaining perturbations for training.

The attached codebase has a python notebook responsible for generating this split: `notebooks/neurips2025/build_jiang24_frangieh21_splits.ipynb`.

**OP3 splits** We reuse the original train, public test and private test data partitions from the NeurIPS 2023 perturbation prediction challenge, in order to create our own train, validation and test splits. The negative controls, DMSO, are used as control cells, and are allocated at a 50/25/25 ratio to train, validation and test splits. The two positive controls, Belinostat and Dabrafenib, are assigned to the training split. The validation and test splits are filtered to remove perturbation and cell type combinations that contain less than 100 perturbed cells.

The attached codebase has a python notebook responsible for generating this split: `notebooks/neurips2025/build_op3_splits.ipynb`.

**Srivatsan20 Data Imbalance splits** To generate the imbalanced `Srivatsan20` datasets for Figure C.1, we set three different desired level of imbalance, which we have quantified via normalized entropy based on the number of perturbations per cell line:

$$\text{Imbalance} := 1 - \frac{\sum_{i=1}^k \frac{n_i}{n} \log \frac{n_i}{n}}{\log k},$$

where $n_i, i = 1, \ldots, k$ denotes the number of samples in class $i$ and $n = \sum_{i=1}^n n_i$ the overall number of observations. The full `Srivatsan20` data is fully balanced with 188 perturbations seen in all three cell lines. For the three subsequent imbalanced data sets, we fix the first cell line to always see all 188 perturbations, and then randomly choose the number of seen perturbations for the other two cell lines that will result in the desired level of balance (distributions given in Table 14). Control cells are

Table 14: Number of perturbations in each cell line for downsampled subsets of `Srivatsan20` with different levels of data balance.

| Balance | # Perts Cell Line 1 | # Perts Cell Line 2 | # Perts Cell Line 3 |
|---|---|---|---|
| 1 | 188 | 188 | 188 |
| 0.9 | 188 | 50 | 117 |
| 0.8 | 188 | 81 | 30 |
| 0.7 | 188 | 33 | 33 |

always seen in training for each cell line. We then randomly downsample each cell line to the desired number of perturbations, and use our datasplitter with default parameters to generate a cross cell line split. We set the minimum number of perturbations to 30 per cell line.

**Unseen perturbation splits**   Some models such as scGPT, GEARS, and PerturbNet create an embedding over the perturbation space which enables prediction of the effect of perturbations that are never seen during training in any context. Since this task is very complex and most likely highly dependent on the quality of the perturbation embedding/representation, we choose not to address it the scope of this study.

### D.5   Models

**CPA**   We implement a version of CPA using the published Theis lab model (forked 02/23). The Theis lab codebase has been updated since publication, and we have incorporated the most important changes into our implementation. However, some small differences remain, thus, we refer to our implementation as CPA$^*$.

To ensure that we have correctly implemented CPA, we verify that our implementation has achieved similar or better performance on all metrics compared to the published versions. To this end, we have trained a CPA model using the published Theis lab model (forked 02/23) and our implementation using the exact same hyperparameters identified to be optimal by the authors, on the same data split and assessed the performance. Our implementation of CPA has obtained comparable (and indeed, slightly better) results than the original codebase. CPA alternates between training the generator and discriminator, and in the ablated version: CPA$^*$ (noAdv), we disregard the discriminator by setting the adversarial loss to zero and training the generator exclusively.

**SAMS-VAE**   SAMS-VAE is available under a restrictive licence. For this reason, we implement a version of the model carefully following the authors' description. Since, the model is re-implementation, we refer to it as SAMS-VAE$^*$.

To further understand the effect of sparse additive mechanism, we ablate the sparsity-inducing component of SAMS-VAE by completely removing the binary mask and its associated global latent variable. The other significant modification is to remove the global variable defined on the perturbation embedding. To this end, we obtain a simplified version, i.e. SAMS-VAE$^*$ (S), which does not contain any global latent variables, and learns perturbation effects through ordinary embedding vectors without any sparsity or distributional assumptions.

**BioLord**   For modelling the effect of perturbations, Biolord requires embeddings, either from the GEARS GO graph for genes or RDKIT embeddings for small molecules. For the sake of fair comparison, we have excluded the use of embeddings and therefore, implemented a slight variation of Biolord, henceforth referred to as Biolord$^*$, where instead of neighbourhoods based on embeddings, we use the same one hot representation for perturbations as for all other models.

**GEARS**   Because the GEARS model differs from other models in its use of GNNs to encode gene expression values and perturbations, as well as the authors did not recommend applying it to the *covariate transfer* task, we choose not to reimplement GEARS in our library. We instead write a helper function and HPO script for training and evaluating GEARS using its publicly available code, on the same `Norman19` split which we have used for other models.

**scGPT Embeddings**  To generate scGPT embeddings, we use the pretrained whole human model and generate embeddings with no further fine-tuning on our processed datasets.

**Linear**  The simple linear baseline model uses the *control matching* approach. Given a perturbed cell, $x'$, we sample a random control cell with *matched* covariates, $x$, and reconstruct $x'$ by applying one linear layer given the perturbation and covariates:

$$x' = x + f_{\text{linear}}(p_{\text{one\_hot}}, cov_{\text{one\_hot}}), \tag{6}$$

where $p_{\text{one\_hot}}$ denotes the one-hot encoding of the perturbation and $cov_{\text{one\_hot}}$ denotes one-hot encodings of covariates (i.e. cell type/line).

**Latent Additive**  We extended the linear model into a baseline Latent Additive model by encoding expression values and perturbations into a latent space $\mathsf{Z} \subseteq \mathbb{R}^{d_z}$, i.e.

$$z_{\text{ctrl}} = f_{\text{ctrl}}(x), \quad \text{and} \quad z_{\text{pert}} = f_{\text{pert}}(p_{\text{one\_hot}}),$$

where $p_{\text{one\_hot}}$ denotes the one-hot encoding of the perturbation. Subsequently, we reconstruct the expression value by decoding the added latent space representation $x' = f_{\text{dec}}(z_{\text{ctrl}} + z_{\text{pert}})$.

**Decoder Only**  As a further ablation study, we introduce a model class that aims to predict the transcriptome solely from covariates, $cov_{\text{one\_hot}}$, perturbation information, $p_{\text{one\_hot}}$, or a mix of both. This model takes as an input neither the transcriptome of a control cell nor the transcriptome of a perturbed cell. Consequently, prediction of the expression of a perturbed cell can be modelled as $x' = f_{\text{dec}}(z)$ for $z \in \{p_{\text{one\_hot}}\} \cup \{cov_{\text{one\_hot}}\} \cup \{(p_{\text{one\_hot}}, cov_{\text{one\_hot}})\}$ and we refer to them as *Decoder-Only* models. This class of models provides a range of baselines:

- Firstly, a model decoding only from covariates provides a lower bound on the performance of acceptable models and a sense of what performance can be expected when a model collapses to its mode(s). For instance, if the covariates contain only the cell type/line, this model will only learn the average expression value for each cell type/line. Since no perturbation information is used, the model is completely collapsed for every class of covariates.

- Secondly, a model that decodes only from perturbations offers a baseline that illustrates the extent to which expression levels resulting from perturbations can be predicted, disregarding any information about cell type/line or expression levels in control cells.

- Thirdly, a model that decodes information from both cell type/line and perturbations provides a baseline for understanding the additional information that the transcriptome could offer, which is not already captured by the covariates or inherently present in the perturbation data.

### D.6  Hyperparameter Optimization

#### D.6.1  Identifying a Hyperparameter Metric

In order to carry out HPO, we need to define a performance metric that can be taken as an objective function for `optuna`. The model loss calculated on the validation data can in many cases be unsuitable for such a task, as some hyperparameters are part of the loss itself and aim, for instance, to find a balancing factor between different loss terms. In such scenarios, the objective would induce `optuna` to simply set a scaling factor to 0. Hence, we require an alternative metric as an HPO objective function.

To define an objective functions we set out the following requirements:

- To make our models comparable and to avoid confounding issues, we compare all models based on the same metric for the purposes of HPO.

- Considering the results of Section C.7 hyperparameter optimization can not simply be carried out on one metric, such as RMSE, as we have established that this metric alone does not cover all aspects of model performance.

To identify suitable hyperparameter metrics, we carried out several HPO runs with linear combinations of cosine similarity and the respective rank metric, as well as RMSE and its respective rank metric. In a few pilot hpo runs we observed that

$$\mathcal{L}_{\text{HPO}} = \text{RMSE} + 0.1 \cdot \text{rank}_{\text{RMSE}}$$

results in models that perform well on both aspects, traditional model fit as well as ranking metrics.

### D.6.2 Hyperparameter Ranges

For hyperparameter optimization we used `optuna` [4]. Hence, we can define all hyperparameter ranges as `optuna` distributions, either in the form of `categorical`, `int` or `float`. We describe the seed and the specific `optuna` hyperparameter ranges as well as their distribution classes in Tables 15 to 19 and 21.

Table 15: CPA hyperparamter range.

| Hyperparameter | Distribution |
|---|---|
| *Number of layers in the encoder part of the model:* | |
| `n_layers_encoder` | `Int:  1 to 7, step=2` |
| *Number of perturbation embedding layers:* | |
| `n_layers_pert_emb` | `Int:  1 to 5, step=1` |
| *Number of layers in the adversarial classifier:* | |
| `adv_classifier_n_layers` | `Int:  1 to 5, step=1` |
| *Hidden dimension size:* | |
| `hidden_dim` | `Int:  256 to 5376, step=1024` |
| *Hidden dimension size of the adversarial classifier:* | |
| `adv_classifier_hidden_dim` | `Int:  128 to 1024, log=True` |
| *Number of adversarial steps:* | |
| `adv_steps` | `Categorical:  [2, 3, 5, 7, 10, 20, 30]` |
| *Number of latent variables:* | |
| `n_latent` | `Categorical:  [64, 128, 192, 256, 512]` |
| *Learning rate:* | |
| `lr` | `Float:  5e-6 to 1e-3, log=True` |
| *Weight decay:* | |
| `wd` | `Float:  1e-8 to 1e-3, log=True` |
| *Dropout rate:* | |
| `dropout` | `Float:  0.0 to 0.8, step=0.1` |
| *KL divergence weight:* | |
| `kl_weight` | `Float:  0.1 to 20, log=True` |
| *Adversarial weight:* | |
| `adv_weight` | `Float:  0.1 to 20, log=True` |
| *Penalty weight:* | |
| `penalty_weight` | `Float:  0.1 to 20, log=True` |

Table 16: Latent additive model hyperparameter range.

| Hyperparameter | Distribution |
|---|---|
| *Number of layers in the encoder part of the model:*
n_layers | Int:  1 to 7, step=2 |
| *Width of the encoder layers in the model:*
encoder_width | Int:  256 to 5376, step=1024 |
| *Dimensionality of the latent space:*
latent_dim | Categorical:  [64, 128, 192, 256, 512] |
| *Learning rate:*
lr | Float:  5e-6 to 5e-3, log=True |
| *Weight decay:*
wd | Float:  1e-8 to 1e-3, log=True |
| *Dropout rate:*
dropout | Float:  0.0 to 0.8, step=0.1 |

Table 17: Linear additive model hyperparameter range.

| Hyperparameter | Distribution |
|---|---|
| *Learning rate:*
lr | Float:  5e-6 to 5e-3, log=True |
| *Weight decay:*
wd | Float:  1e-8 to 1e-3, log=True |

Table 18: Biolord hyperparameter range.

| Hyperparameter | Distribution |
|---|---|
| *Weight of the penalty term in the loss function:*
penalty_weight | Float:  1e1 to 1e5, log=True |
| *Number of layers in the encoder part of the model:*
n_layers | Int:  1 to 7, step=2 |
| *Width of the encoder layers in the model:*
encoder_width | Int:  256 to 5376, step=1024 |
| *Dimensionality of the latent space:*
latent_dim | Categorical:  [64, 128, 192, 256, 512] |
| *Learning rate:*
lr | Float:  5e-6 to 5e-3, log=True |
| *Weight decay:*
wd | Float:  1e-8 to 1e-3, log=True |
| *Dropout rate:*
dropout | Float:  0.0 to 0.8, step=0.1 |

Table 19: SamsVae hyperparameter range.

| Hyperparameter | Distribution |
|---|---|
| *Number of layers in the encoder part of the model:* | |
| n_layers_encoder_x | Int: 1 to 7, step=2 |
| *Number of layers in the encoder part of the model:* | |
| n_layers_encoder_e | Int: 1 to 7, step=2 |
| *Number of layers in the decoder part of the model:* | |
| n_layers_decoder | Int: 1 to 7, step=2 |
| *Width of the encoder layers in the model:* | |
| latent_dim | Categorical: [64, 128, 192, 256, 512] |
| *Hidden dimension for x:* | |
| hidden_dim_x | Int: 256 to 5376, step=1024 |
| *Hidden dimension for the conditional input:* | |
| hidden_dim_cond | Int: 50 to 500, step=50 |
| *Whether to use sparse additive mechanism:* | |
| sparse_additive_mechanism | Categorical: [True, False] |
| *Whether to use mean field encoding:* | |
| mean_field_encoding | Categorical: [True, False] |
| *Learning rate:* | |
| lr | Float: 5e-6 to 1e-3, log=True |
| *Weight decay:* | |
| wd | Float: 1e-8 to 1e-3, log=True |
| *The target probability for the masks:* | |
| mask_prior_probability | Float: 1e-4 to 0.99, log=True |
| *Dropout rate:* | |
| dropout | Float: 0.0 to 0.8, step=0.1 |

Table 20: DecoderOnly hyperparameter range.

| Hyperparameter | Distribution |
|---|---|
| *Number of layers in encoder/decoder:* | |
| n_layers | Int: 1 to 7, step=2 |
| *Width of the encoder:* | |
| encoder_width | Int: 256 to 5376, step=1024 |
| *Learning rate:* | |
| lr | Float: 5e-6 to 5e-3, log=True |
| *Weight decay:* | |
| wd | Float: 1e-8 to 1e-3, log=True |
| *Whether to apply a softplus activation to the output of the decoder to enforce non-negativity:* | |
| softplus_output | Categorical: [True, False] |

Table 21: GEARS hyperparameter range.

| Hyperparameter | Distribution |
| --- | --- |
| *Number of layers in perturbation GNN:* 
 num_go_gnn_layers | Int: 1 to 3, , step=1 |
| *Number of layers in gene GNN:* 
 num_gene_gnn_layers | Int: 1 to 3, step=1 |
| *Number of neighboring perturbations in GO graph:* 
 num_similar_genes_go_graph | Int: 10 to 30, step=10 |
| *Number of neighboring genes in gene co-expression graph:* 
 num_similar_genes_co_express_graph | Int: 10 to 30, step=10 |
| *Width of the encoder:* 
 hidden_size | Int: 32 to 512, step=32 |
| *Minimum coexpression threshold:* 
 co_express_threshold_graph | Float: 0.2 to 0.5, step=0.1 |
| *Learning rate:* 
 lr | Float: 5e-6 to 5e-3, log=True |
| *Weight decay:* 
 wd | Float: 1e-8 to 1e-3, log=True |

### D.7 Compute Resources

For the `Norman19` and `Srivatsan20`, and data imbalance tasks, we used nodes with one Nvidia A10G GPU each. We ran 60 hyperparameter optimization trials for each model, and assessed 10 models on the `Srivatsan20` task and 9 models on the `Norman19` task. We also ran 4 training runs with the best hyperparameters for stability analysis. We also ran an additional 5 models on the 4 different data imbalance splits, again with 4 runs for stability. For the HPO runs we used 813 hours for `Srivatsan20` and 399 hours for `Norman19`. See details in Table 22.

For the `McFalineFigueroa23` data scaling task, we used nodes with Nvidia A10G GPUs for most of the combinations of models and subsets. We used A100G or H100G GPUs for all deep learning model for the biggest split, and for all datasets on CPA (which required the most GPU memory). We again used 60 hyperparameter optimization trials across 4 models with an additional 4 runs for stability. In total for this experiments we used 2517 hours of servers with GPUs, see details in Table 22.

Table 22: Total runtime of HPO for different models and datasets

| dataset | model | runtime | A100 |
|---|---|---|---|
| mcfaline23-full | cpa | 171.97 | yes |
| mcfaline23-full | decoder-only | 136.91 | yes |
| mcfaline23-full | latent-additive | 150.36 | yes |
| mcfaline23-full | linear-additive | 321.08 | |
| mcfaline23-medium | cpa | 127.44 | yes |
| mcfaline23-medium | decoder-only | 225.24 | |
| mcfaline23-medium | latent-additive | 280.12 | |
| mcfaline23-medium | linear-additive | 359.33 | |
| mcfaline23-small | cpa | 105.12 | yes |
| mcfaline23-small | decoder-only | 135.38 | |
| mcfaline23-small | latent-additive | 186.14 | |
| mcfaline23-small | linear-additive | 317.91 | |
| norman19 | biolord | 129.71 | |
| norman19 | cpa | 42.98 | |
| norman19 | cpa-no-adversary | 48.08 | |
| norman19 | cpa-scgpt | 25.20 | |
| norman19 | decoder | 20.48 | |
| norman19 | latent | 32.69 | |
| norman19 | latent-scgpt | 21.42 | |
| norman19 | linear | 30.17 | |
| norman19 | sams | 48.06 | |
| sciplex3 | biolord | 312.49 | |
| sciplex3 | cpa | 41.66 | |
| sciplex3 | cpa-no-adversary | 51.83 | |
| sciplex3 | cpa-scgpt | 38.54 | |
| sciplex3 | decoder | 40.41 | |
| sciplex3 | decoder-cov | 36.42 | |
| sciplex3 | latent | 56.59 | |
| sciplex3 | latent-scgpt | 76.25 | |
| sciplex3 | linear | 70.48 | |
| sciplex3 | sams | 88.76 | |

