# OpenReview forum: "PerturBench: Benchmarking Machine Learning Models for Cellular Perturbation Analysis"
_NeurIPS.cc/2025/Datasets_and_Benchmarks_Track — NeurIPS 2025 Datasets and Benchmarks Track poster_

### Official Review · Reviewer_zMhV · 2025-06-26

**Rating:** 6
**Confidence:** 3

**Summary:**

This paper introduces PerturBench, a modular and comprehensive benchmarking framework for evaluating machine learning models in the context of single-cell perturbation response modeling. It incorporates a library of diverse perturbation datasets, a suite of biologically-relevant tasks (e.g., covariate transfer and combo prediction), a set of complementary metrics (including novel rank-based metrics), and a useful framework for model development. The authors evaluate a wide range of existing models and new baselines, offering rigorous comparative insights across datasets of varying size and complexity.

**Additional Feedback:**

A few suggestions for improvement:
The authors could check if the top-ranked gene or drug perturbations are already known to be important in a disease, or if the predicted changes in gene expression match known biological pathways.

You may also include direct comparisons with original implementations of CPA, SAMS-VAE, etc., or provide side-by-side reference values.

Also, you should either include the appendices in the main pdf, or discard the links to them.
Please add a paragraph regarding ethical considerations.

**Dataset Code Accessibility:**

Yes

**Dataset Code Comments:**

The code is available on github and the data in HuggingFace. The documentation on github clearly explains how to use the benchmarking tools and dataset.

**Ethical Considerations:**

No, there are no or only very minor ethics concerns

**Final Justification:**

The authors have addressed all of my concerns and promised to add those changes in the camera ready version of their paper.

In particular:
They explained to me that they were not able to conduct lab experiments, but used published real-world datasets

They also provided clearer details on the implementation differences with the baselines (CPA, SAMS-VAE, etc.)

Geneformer aims to solve a different task (order of the genes) rather than actual expression values of the genes.

Moreover, they aim to add an ethical statement in their next version of the paper.

**Limitations Weaknesses:**

My main concerns are regarding the benchmarking experiments.
The benchmarking would be strengthened by a real-world application or validation. For instance, experimentally confirming high-ranking perturbations in a disease model would help assess whether the proposed metrics truly correlate with biological relevance.

As a minor weakness, the paper acknowledges that the models implemented (CPA, SAMS-VAE, etc.) may deviate from the original implementations. While this is reasonable given the benchmarking scope, clearer detail on which parts differ and how they affect performance would strengthen the claims.

You can also enrich your benchmarking with additional well known models like Geneformer.

**Strengths Contributions:**

The paper addresses a need in the field where inconsistent evaluation protocols have hindered progress.

The proposed benchmark covers five major single-cell perturbation datasets, representing different perturbation modalities (chemical/genetic), biological contexts, and dataset sizes. Tasks like covariate transfer and combo prediction are particularly relevant to real-world applications.

Also, rank-based metrics better capture a model's utility in downstream prioritization tasks and are particularly useful for in-silico screening applications. The intuition behind the metric is well explained, and its ability to diagnose mode/posterior collapse is compelling.

The open-source codebase is modular and extensible, making it easy for researchers to add new models or datasets. This helps build a standardized benchmark for the field.

---

> ### Author Rebuttal · Authors · 2025-07-31
>
> We would like to thank you for the insightful and positive review of our work.
>
> > My main concerns are regarding the benchmarking experiments. The benchmarking would be strengthened by a real-world application or validation. For instance, experimentally confirming high-ranking perturbations in a disease model would help assess whether the proposed metrics truly correlate with biological relevance.
>
>
> We acknowledge that the work could be strengthened by a real-world application or validation and appreciate the comment. Unfortunately, we were not able to commit wet lab resources to the publication and thus we had to rely on previously published open datasets.
>
> Moreover, we focused on evaluating prediction of transcriptomes after perturbation, which we hope is a universal and useful task. In order to select for the “high-ranking perturbations in a disease model”, one would also need to develop methods to rank desirable transcriptomes for specific biological applications. We agree that these ranking methods would be incredibly useful, but we feel they are beyond the scope of PerturBench.
>
> > A few suggestions for improvement: The authors could check if the top-ranked gene or drug perturbations are already known to be important in a disease, or if the predicted changes in gene expression match known biological pathways.
>
> Thank you for the suggestion. Analysing the “top-ranked gene or drug” has the aforementioned challenge of defining the ranking, which we consider to be out of the scope. The comparison to known biological pathways is a very interesting idea, but may be challenging to identify whether given pathways make sense for a perturbation response where we are evaluating the responses of hundreds of perturbations in each cell type, across dozens of cell types and multiple datasets. This type of analysis may become feasible with the ongoing development of AI agents, and we see this as an exciting area for future research.
>
> We also acknowledge that an analysis for a limited number of perturbations could still give an interesting qualitative insight about model performance, but it is not something we were able to conduct during the limited time of rebuttal period.
>
> > You can also enrich your benchmarking with additional well known models like Geneformer.
>
> We acknowledge the fact that the benchmark would benefit from more models and we hope it will be expanded over time, especially as it is already getting some attention in the community. On incorporating the Geneformer model that you suggested, the challenge is that it is predicting the order of the gene by their expression, not actual values of the expression, which makes it incompatible with most of the metrics we have introduced.
>
> > As a minor weakness, the paper acknowledges that the models implemented (CPA, SAMS-VAE, etc.) may deviate from the original implementations. While this is reasonable given the benchmarking scope, clearer detail on which parts differ and how they affect performance would strengthen the claims.
>
> In order to allow easy experimentation with the models, in particular to be able to easily verify the impact of their different components, we decided to reimplement the models in a common codebase. The details of differences between our implementations and reference are summarized in the Appendix D.5. Please let us know, if there are aspects of the models that require further explanation. The users can also verify the details of implementation in the open source codebase that we released.
>
> > You may also include direct comparisons with original implementations of CPA, SAMS-VAE, etc., or provide side-by-side reference values.
>
> Thank you for the suggestion. To confirm that our implementation of CPA is indeed correct, we verified that our CPA implementation matches or exceeds the published CPA implementation performance on the Norman19 dataset, where CPA_Theis is the public version of CPA hosted on github and accessed on 07/30/2025, and CPA_perturbench is our recreation of CPA. We specifically trained a CPA_Theis model using their Norman19 example notebook with provided hyperparameters and their version of the Norman19 dataset, using the “split_6” train/val/test split. We then added their version of the Norman19 dataset to PerturBench, and trained our CPA implementation on the same dataset/split. We evaluated both model’s performance on the held out “ood” split using PerturBench metrics. We plan on adding these verification notebooks and config files to the PerturBench repo.
>
> |                         |   CPA_Theis |   CPA_perturbench |
> |:------------------------|------------:|-------------:|
> | rmse_average            |     0.1124  |     0.02729  |
> | rmse_rank_average       |     0.4267  |     0.008889 |
> | cosine_logfc            |     0.2058  |     0.7498   |
> | cosine_rank_logfc       |     0.2622  |     0.01333  |
> | r2_score_scores         |    -1.004   |     0.2247   |
> | top_k_recall_scores     |     0.09867 |     0.384    |
> | mmd_pca                 |     3.283   |     2.308    |
> | mmd_rank_pca            |     0.4222  |     0.008889 |
> | mmd_none                |     4.237   |     3.769    |
> | mmd_rank_none           |     0.4133  |     0.008889 |
>
> We noticed that the choice of loss has a major impact on performance, and models that used a simple MSE loss performed better than Gaussian or Negative Binomial Negative Log Likelihood (NLL) losses. We chose to use the MSE loss in our CPA reproduction, whereas the public model uses NLL losses we believe is the primary reason why our reimplemented CPA performs significantly better than the public CPA implementation. We feel that the use of MSE loss enables us to compare the rest of the model components more on a more even basis, but can add our findings on the NLL vs MSE loss to the manuscript as well.
>
> It is important to note that due to the license of SAMS-VAE codebase, we were not able to run that model and performed a head-to-head comparison.
>
> > Also, you should either include the appendices in the main pdf, or discard the links to them. Please add a paragraph regarding ethical considerations.
>
> Thank you very much for both suggestions, we will definitely be very happy to address them in the camera ready version, as it is not possible to edit the manuscript during the rebuttal period.

---

> > ### Author Response · Authors · 2025-08-05
> >
> > Dear reviewer zMhv,
> >
> > We have replied to your concerns and comments above. Please let us know if there are any additional concerns from you that we could help address. Thank you.

---

> > > ### Comment · Reviewer_zMhV · 2025-08-06
> > > **Thank you**
> > >
> > > I thank the authors for their detailed response, which has addressed all of my concerns. In light of these changes I am raising my score and hoping that the authors will incorporating the changes in the paper.

---

### Official Review · Reviewer_TLxV · 2025-06-29

**Rating:** 4
**Confidence:** 5

**Summary:**

It is a benchmark for the single-cell perturbation transcriptomic outcome prediction task. It covers the tasks, chemical, genetic, and a combination of genetic perturbations. It introduces a rank-based metric and, in addition to published methods, evaluates a few baselines. The evaluation is performed on 4 genetic and 1 chemical cell line dataset. The benchmark evaluates models using different training data sizes and illustrates the issue of mode collapse.

**Dataset Code Accessibility:**

Yes

**Dataset Code Comments:**

Reproducible code is provided on github, and the dataset is uploaded to huggingface.

**Ethical Considerations:**

No, there are no or only very minor ethics concerns

**Final Justification:**

The authors engaged in addressing some of the limitations of the work. Evaluation on a single-cell dataset and the use of distributional metrics has revealed novel insights, and I believe it would be best if these were studied and incorporated before resubmission; however, I have raised my score significantly.

**Limitations Weaknesses:**

1. It is called a "perturbation response modeling in single cells" benchmark, while it only does benchmarking of transcriptomic effects of perturbations, and only uses pseudobulk metrics on cell lines, effectively making the task more suitable for the evaluation of transcriptomic effects in bulk or pseudobulk samples rather than in single cells. For such a task, many other methods have been proposed that are not cited or mentioned here[1, 2, 3].
2. Evaluation is only performed on cell line datasets even though biologically and therapeutically relevant models, especially for single-cell studies, are available and increasing in numbers [4,5,6]
3. The results show that "simple architectures are generally competitive", which has been shown multiple times before in the literature [5, 7]
4. The definition of "covariate transfer", especially within the context of the benchmark, could be clarified - it is unclear from it whether the covariates or perturbations or only pairs (covariate, perturbation) are unseen. This is important from a modeling perspective since models that need to generalize to unseen covariates or perturbations need to use some embeddings or priors of these, while models that generalize only to unseen pairs don't.
5. The paper calls cell lines cell types e.g. in line 217, which is confusing two distinct concepts.
6. The paper misrepresents prior work:
a. In line 36 the paper of the NeurIPS23 challenge isn't cited[5], where [5] uses a set of metrics unlike claimed in line 38.
b. In line 41 the claim that "Ahlmann-Eltze et al. [2], Wenteler et al. [56], Csendes et al. [12], and Wong et al. [58]" assessed only the performance of single-cell foundation models (umbrella term) and GEARS is incorrect - models like CPA and numerous baselines are evaluated in them.
c. similarly in line 42 claim about only MSE being used as a metric is untrue - for example Ahlmann-Eltze used Pearson delta as well.
d. claim in line 45 about missing "strong deep learning baselines" is vague and given numerous models being evaluated in prior literature, it is hard to agree with it.
e. similarly in the same line "limited set of metrics".
f. model development framework is claimed as novelty, but it is not elaborated how it improves on existing benchmarking frameworks like [5].
[1] Tong, Xiaochu, et al. "Deep representation learning of chemical-induced transcriptional profile for phenotype-based drug discovery." Nature Communications 15.1 (2024): 5378.
[2] Pham, Thai-Hoang, et al. "A deep learning framework for high-throughput mechanism-driven phenotype compound screening and its application to COVID-19 drug repurposing." Nature machine intelligence 3.3 (2021): 247-257.
[3] Zhan, Lingmin, et al. "A genome-scale deep learning model to predict gene expression changes of genetic perturbations from multiplex biological networks." Briefings in Bioinformatics 25.5 (2024): bbae433.
[4] Jiang, Jialong, et al. "D-SPIN constructs gene regulatory network models from multiplexed scRNA-seq data revealing organizing principles of cellular perturbation response." BioRxiv (2024): 2023-04.
[5] Szałata, Artur, et al. "A benchmark for prediction of transcriptomic responses to chemical perturbations across cell types." Advances in Neural Information Processing Systems 37 (2024): 20566-20616.
[6] https://www.parsebiosciences.com/datasets/10-million-human-pbmcs-in-a-single-experiment/#download
[7] Csendes, Gerold, et al. "Benchmarking foundation cell models for post-perturbation RNA-seq prediction." BMC genomics 26.1 (2025): 393.

**Strengths Contributions:**

1. The paper is well-organized and easy to understand
2. It introduces a novel way to evaluate models, rank metrics
3. It includes some model component ablations

---

> ### Author Rebuttal · Authors · 2025-07-31
>
> Thank you for the detailed review of our work. We are glad that you acknowledge the novelty of our rank metric, liked the model component ablation and found the paper well-organized. Below we address the concerns you have raised in order.
>
> To avoid the confusion, when referring to publications number in italics (e.g. *[5]*) we are referencing the list that you provided in your review. Without italics, e.g. Wong et al. [58], we are referring to a publication referenced by our manuscript.
>
> > 1. Data modality and evaluation metrics
>
> Thank you for the comment, we will update the abstract to “We introduce a comprehensive framework for perturbation response modeling in single cell transcriptomic data”, and clarify the scope of our work in the introduction. And thank you for the additional citations. Our benchmark is focused on single-cell perturbation response modeling and we will report additional distributional metrics for single cells.
>
> PerturBench already supports single cell metrics such as maximum mean discrepancy (MMD) and differentially expressed gene (DEG) recall. We now report them for the Jiang24 dataset. Results for the Norman19 and Srivatsan20 dataset can be found in the response to reviewer 2oW4 and were not included here due to the character limit.
>
> To briefly introduce these metrics:
>
> * MMD computes the energy distance between predicted cells and the ground truth perturbed cells in the gene expression (MMD GEX) or PCA (MMD PCA) space. We use the geomloss MMD implementation with the energy kernel. To compute PCs, we fit a PCA model to the ground truth test split, and project predicted cells onto the ground truth PCs. We also compute the rank versions of these metrics, MMD GEX rank and MMD PCA rank.
>
> * DEG recall measures the fraction of ground truth DEGs that can be recalled from the predicted cells. We computed t-scores using the scanpy.tl.rank_genes_groups method, and compared them using top k recall. Since the DecoderOnly model has a variance of zero, we set its DEG recall to N/A.
>
> **Jiang24**
> | model             | MMD GEX     | MMD PCA      | MMD GEX rank   | MMD PCA rank   | DEG recall      |
> |:------------------|:------------|:-------------|:---------------|:---------------|:----------------|
> | CPA               | 8.0 ± 2E-03 | 2.5 ± 4E-03  | 0.44 ± 7E-03   | 0.43 ± 4E-03   | 0.0044 ± 2E-03  |
> | CPA noAdv         | 8.0 ± 2E-03 | 2.5 ± 3E-03  | 0.44 ± 6E-03   | 0.42 ± 4E-03   | 0.0052 ± 6E-04  |
> | SAMS-VAE          | 4.5 ± 7E-03 | 0.32 ± 5E-03 | 0.47 ± 3E-03   | 0.43 ± 1E-02   | 1.8e-05 ± 4E-05 |
> | SAMS-VAE improved | 6.5 ± 2E+00 | 1.6 ± 1E+00  | 0.44 ± 2E-02   | 0.43 ± 1E-02   | 0.0015 ± 1E-04  |
> | Latent            | 8.0 ± 9E-04 | 2.6 ± 2E-03  | 0.41 ± 5E-03   | 0.40 ± 5E-03   | 0.0010 ± 5E-04  |
> | Decoder           | 8.0 ± 2E-03 | 2.6 ± 2E-03  | 0.36 ± 9E-03   | 0.38 ± 8E-03   | N/A   |
> | Linear            | 3.1 ± 2E-03 | 1.3 ± 2E-03  | 0.44 ± 1E-03   | 0.45 ± 6E-04   | 0.0033 ± 2E-04  |
>
>
>
> The MMD and DEG recall metrics assess the full distribution of cellular responses to perturbation. The SAMS-VAE and CPA reimplementations perform better on MMD PCA than the baseline Latent and Decoder models, while the baseline models still outperform the published models on the MMD PCA rank metrics.
>
> All models performed poorly on DEG recall, suggesting that they struggle to predict differential expression t-scores. Models also performed worse on MMD GEX compared to MMD PCA, suggesting that predicting gene expression distributions is still a major challenge. SAMS-VAE and CPA still outperformed the baseline Latent and Decoder models for MMD GEX.
>
> We plan to add the MMD and DEG recall metrics for all datasets to the manuscript.
>
> > 2. Dataset limitation
>
> We plan to add the Open Problems Perturbation Prediction (OP3) benchmark dataset from *[5]* that includes chemical perturbations in primary human PBMCs. We also plan on adding an in-vivo perturb-seq dataset from Liu et al, Genome Biology, 2024, that assesses how well these perturbation models can generalize from in-vitro to in-vivo settings.
>
> Our current benchmark does include the Frangieh2021 dataset, which contains genetic perturbations in primary melanoma cells in co-culture with PBMCs and represents a more complex cell model that captures the immune - tumor interactions in cancer.
>
> We also offer an easily extensible codebase that allows users to add datasets by simply adding a config file. For example, the first author of *[5]* has leveraged our work by forking a version of PerturBench and adding the OP3 dataset and its config.
>
>
>
> > 3. Findings on simple architectures performance
>
> We recognize that multiple prior works have used simple baseline models such as mean predictors and linear models. However, in the scope of our work, we think that the simple deep learning baselines (Latent Additive, Decoder Only), and especially their competitive performance, help us understand what factors affect model performance. In fact, studies such as Wong et al. [58] and *[5]* have used our baseline Latent Additive model. Thus, we feel that our contribution here is in comparing linear baseline, as well as deep learning baselines with published models such as CPA and SAMS-VAE, across diverse datasets, evaluated with our novel rank metrics.
>
> > 4. Clarification on “covariate transfer”
>
> We include a description of how the datasets are split into train/validation/test in both the main text and the methods. We thank the reviewer for pointing out the ambiguity and will add additional clarification (line 31) that “covariate transfer” refers only to unseen pairs.
>
> > 5. Cell lines and cell types confusion
>
> Thank you for noting the inconsistency, we will clarify that these are cell lines in the text.
>
> > 6. Prior work misrepresentation.
>
> Thank you for noting the issues with our prior work. We want to clarify that we did not attempt to misrepresent any prior studies and that your feedback has been crucial for improving the accuracy of our literature review. We provide an updated prior work below and would appreciate any additional feedback.
>
> Comparing the performance of published models has been challenging due to inconsistent benchmarks with different datasets and metrics. The NeurIPS 2023 perturbation prediction challenge described in *Szałata et al. [5]*, was a major achievement in standardizing benchmarks, providing a novel chemical perturbation dataset with scRNA-seq readouts measured in PBMCs. The challenge used the covariate transfer task, with metrics including mean row-wise RMSE, MAE, and cosine similarity of predicted vs ground truth log p-values. The challenge attracted a large number of submissions, many of which were inspired by published models such as chemCPA [5].
>
> The sc-perturb database provides datasets with unified metadata, but does not attempt to benchmark models [41].
>
> Ahlmann-Eltze et al. [2], Wenteler et al. [56], Csendes et al. [12], and Wong et al. [58] evaluate single-cell foundation models (scFM) such as scGPT, scFoundation, scBERT, Geneformer and UCE, in the context of perturbation response modeling. These studies focus on how these general-purpose models can be fine-tuned for this task, using task-specific models such as GEARS, CPA as well as other baselines (e.g., mean prediction, kNN, random forest, linear models) to highlight the limitations of scFMs for this task. Notably, Wong et al. [58] has used our baseline models and rank metric to establish the performance of scGPT and GEARS.
>
> These works mostly use well-known model fit metrics such as RMSE/MSE and Pearson correlation between averaged predicted and ground truth expression profiles (Pearson Delta, Pearson LogFC). Wenteler et al. [56] also proposed various distributional metrics, including E-distance, which is equivalent to our energy distance based MMD metric..
>
> Kernfeld et al. [28] systematically assessed a wide array of perturbation response prediction models, with a focus on models that use gene regulatory networks as a form of prior knowledge. A central finding of their work was that simple baselines often matched or outperformed more sophisticated models such as GEARS and Geneformer, which confirm the robust performance of simple approaches in this domain.
>
> Recent works by C. Li et al. [31] and L. Li et al. [32] have provided more comprehensive benchmarks of a large set of deep learning models across diverse datasets and metrics. Beyond conventional evaluation, their work introduces novel tasks, such as unseen perturbation/covariate transfer [31, 32] and cell state transition prediction [31]. Notably, while scFMs can excel on unseen perturbation prediction, simpler models often show better performance in the unseen covariate prediction [31].
>
> Despite the emergence of various benchmarking efforts, there remains an opportunity for a modular, comprehensive model development and benchmarking platform that 1) enables ablation studies, 2) adds biologically relevant metrics, and 3) contains strong deep learning baselines. This platform would help the field build and systematically understand perturbation response prediction models.
>
> > 7. Model development framework novelty
>
> Thank you for the comment, we hope to clarify the contribution of our benchmarking framework below and add this clarification to the manuscript.
>
> One of the main contributions is that in addition to providing a unified benchmarking framework, PerturBench also offers a model development platform, which is as easy as writing a Pytorch Lightning LightningModule file.
>
> We feel that combining model development, datasets, and benchmarks in a single repository adds unique value compared to other works such as *[5]*, [31], and [32]. For example, PerturBench enabled us to perform ablation studies on CPA and SAMS-VAE that demonstrated that the adversarial component of CPA and the sparsity inducing component of SAMS-VAE were actually detrimental to model performance.

---

> > ### Comment · Reviewer_TLxV · 2025-08-01
> >
> > Thank you for a thoughtful answer.
> > I will increase the score after the authors include distributional metrics in Frangieh2021 and another primary tissue dataset, such as [5], as suggested in the response. This would significantly strengthen the case for this benchmark, highlighting aspects relevant for single-cell perturbation prediction.
> > I appreciate the evaluation on MMD in Jiang24; however, its relevance is limited due to this dataset being limited to cell lines.

---

> > > ### Author Response · Authors · 2025-08-01
> > > **Frangieh21 MMD Metrics**
> > >
> > > Thank you for the quick response and the additional comments, and we especially appreciate your willingness to increase the score. We've added the MMD and DEG recall metrics for the Frangieh21 dataset below. We are currently running hyperparameter optimization on the dataset from [5] and hope to have results by Monday or Tuesday.
> > >
> > > **Frangieh21**
> > > | model             | MMD GEX     | MMD PCA      | MMD GEX rank   | MMD PCA rank   | DEG recall      |
> > > |:------------------|:------------|:-------------|:---------------|:---------------|:----------------|
> > > | CPA               | 2.1 ± 2E-02 | 0.26 ± 5E-03 | 0.44 ± 5E-03   | 0.34 ± 6E-03   | 0 ± 0           |
> > > | CPA noAdv         | 1.8 ± 1E-02 | 0.21 ± 2E-03 | 0.42 ± 4E-03   | 0.25 ± 1E-02   | 0 ± 0           |
> > > | SAMS-VAE          | 2.2 ± 1E-02 | 0.29 ± 2E-03 | 0.49 ± 4E-03   | 0.49 ± 2E-02   | 0 ± 0           |
> > > | SAMS-VAE improved | 2.4 ± 1E-02 | 0.32 ± 4E-03 | 0.41 ± 4E-03   | 0.26 ± 1E-02   | 0 ± 0           |
> > > | Biolord           | 1.1 ± 2E-02 | 0.20 ± 3E-03 | 0.43 ± 4E-03   | 0.29 ± 2E-02   | 4.6e-05 ± 1E-04 |
> > > | Latent            | 6.4 ± 3E-02 | 3.1 ± 3E-02  | 0.24 ± 8E-03   | 0.23 ± 1E-02   | 0 ± 0           |
> > > | Decoder           | 6.6 ± 8E-04 | 3.3 ± 4E-04  | 0.19 ± 3E-03   | 0.17 ± 3E-03   | 0 ± 0           |
> > > | Linear            | 2.1 ± 8E-03 | 0.79 ± 6E-03 | 0.34 ± 7E-04   | 0.34 ± 2E-03   | 0 ± 0           |
> > >
> > > We see that all models struggle with DEG recall on this task. Our reimplementations of the published CPA, SAMS-VAE, and BioLord models outperformed the Linear, Latent, and Decoder baselines in predicting the full distribution of perturbation effects in both PCA and gene expression (GEX) space, as measured by MMD. All models performed significantly better in PCA space compared to GEX space. As with the Jiang24 and other datasets, the baseline Latent and Decoder models still outperform on rank metrics (MMD_GEX_rank, MMD_PCA_rank).

---

> > > > ### Author Response · Authors · 2025-08-04
> > > > **OP3 Benchmarking Results**
> > > >
> > > > We have finished evaluating the models on the OP3 dataset which includes perturbation responses in primary PBMCs. We hope that with the additional distributional metrics, the results for the OP3 dataset, and our revised related work section we have addressed the bulk of your concerns about our manuscript. We kindly ask that you consider raising our score in light of these improvements.
> > > >
> > > > **OP3 Pseudobulk Metrics**
> > > > | model             | Cosine LogFC   | RMSE mean   | Cosine LogFC rank   | RMSE mean rank   |
> > > > |:------------------|:---------------|:---------------|:--------------------|:--------------------|
> > > > | CPA               | 0.42 ± 3E-03   | 0.034 ± 2E-04  | 0.071 ± 4E-03       | 0.048 ± 6E-03       |
> > > > | CPA noAdv         | 0.34 ± 8E-03   | 0.036 ± 5E-04  | 0.057 ± 4E-03       | 0.043 ± 3E-03       |
> > > > | SAMS-VAE          | 0.44 ± 1E-02   | 0.036 ± 7E-04  | 0.067 ± 2E-02       | 0.066 ± 2E-02       |
> > > > | SAMS-VAE improved | 0.46 ± 2E-03   | 0.033 ± 2E-04  | 0.053 ± 4E-03       | 0.040 ± 1E-03       |
> > > > | Latent     | 0.39 ± 2E-02   | 0.036 ± 2E-03  | 0.12 ± 3E-02        | 0.14 ± 3E-02        |
> > > > | Decoder           | 0.32 ± 6E-03   | 0.035 ± 2E-04  | 0.070 ± 5E-03       | 0.068 ± 6E-03       |
> > > > | Linear            | 0.011 ± 9E-05  | 0.070 ± 4E-05  | 0.14 ± 4E-04        | 0.24 ± 9E-04        |
> > > >
> > > > **OP3 MMD and DEG Metrics**
> > > > | model             | MMD GEX    | MMD PCA      | MMD GEX rank   | MMD PCA rank   | DEG Recall   |
> > > > |:------------------|:------------|:-------------|:----------------|:---------------|:----------------------|
> > > > | CPA               | 5.5 ± 3E-02 | 0.81 ± 9E-03 | 0.23 ± 2E-02    | 0.039 ± 8E-03  | 0.0044 ± 5E-04        |
> > > > | CPA noAdv         | 5.8 ± 4E-03 | 0.96 ± 2E-02 | 0.21 ± 1E-02    | 0.036 ± 5E-03  | 0 ± 0                 |
> > > > | SAMS-VAE          | 5.7 ± 4E-02 | 1.0 ± 4E-02  | 0.20 ± 8E-03    | 0.070 ± 1E-02  | 0 ± 0                 |
> > > > | SAMS-VAE improved | 5.6 ± 1E-02 | 0.83 ± 1E-02 | 0.23 ± 1E-02    | 0.044 ± 3E-03  | 0 ± 0                 |
> > > > | Latent     | 11. ± 6E-02 | 4.1 ± 5E-02  | 0.16 ± 2E-02    | 0.13 ± 3E-02   | 2.8e-05 ± 6E-05       |
> > > > | Decoder           | 11. ± 5E-03 | 4.4 ± 1E-02  | 0.077 ± 6E-03   | 0.055 ± 5E-03  | N/A        |
> > > > | Linear            | 4.0 ± 7E-03 | 2.6 ± 4E-03  | 0.12 ± 8E-04    | 0.25 ± 2E-03   | 0 ± 0                 |
> > > >
> > > > Strikingly, we see that both implementations of CPA and SAMS-VAE outperform all the baselines: Latent, Decoder, and Linear models, on both the pseudobulk and distributional metrics in the OP3 dataset task. Ablating the adversarial component of CPA (CPA noAdv) reduces performance on the cosine logFC and RMSE metrics, while slightly improving performance on the rank metrics. Our improved SAMS-VAE implementation which involves ablating the sparse mask performs better on all metrics. The Decoder Only model outperforms the Latent Additive baseline on the rank metrics and is close on the Cosine logFC and RMSE metrics, suggesting that the simpler Decoder Only architecture that maps directly from perturbation to average perturbed cell response is superior for this OP3 dataset when predicting average responses.
> > > >
> > > > All models again perform poorly on the DEG recall task, similar to what we’ve seen for other datasets. The CPA and SAMS-VAE models are significantly better than baselines on the MMD metrics, especially when computed in PCA space (MMD PCA). As with the other datasets, models generally perform worse in the gene expression space (MMD GEX). The MMD metrics in both gene expression and PCA space are worse for the OP3 dataset compared to our other datasets, suggesting a more heterogeneous perturbation response which confirms the value of the added OP3 dataset.

---

### Official Review · Reviewer_jrGj · 2025-06-29

**Rating:** 4
**Confidence:** 4

**Summary:**

This paper compiles a large Perturb-seq dataset from five published studies to investigate gene expression responses to various perturbations at the single-cell level. Perturbations include gene knockouts, overexpression, or chemical/drug treatments. The task is to predict the gene expression profile of perturbed cells based on the perturbation itself and the gene expression profiles of control (untreated) cells. Since scRNA-seq is destructive, the same cells cannot be observed before and after treatment; thus, one must match treated and control cells computationally.

The authors propose a rank-based evaluation metric to measure the similarity between predicted and observed expression profiles. They also reimplement several baseline methods in a modular framework to enable fair comparisons and systematic ablation studies. Interestingly, the benchmark results challenge original claims made in the respective papers: for example, CPA performs better without the adversarial component, and SAMS-VAE performs better without its sparse mask. These findings highlight issues with reproducibility and suggest that simpler architectures may be more effective in practice.

**Dataset Code Accessibility:**

Yes

**Dataset Code Comments:**

Data and code are accessible.

**Ethical Considerations:**

No, there are no or only very minor ethics concerns

**Limitations Weaknesses:**

The claim that the rank-based metric has an expected value of 0.5 under random predictions is not empirically validated. It would strengthen the paper to include experiments using random or naive predictors to confirm this baseline behavior.

The dataset is relatively limited in scope, covering only five publications. A more comprehensive benchmark would benefit from including broader datasets. For instance, PerturBase curates 122 datasets from 46 studies, covering over 24,000 genetic and 230 chemical perturbations from approximately 5 million cells. (see: https://academic.oup.com/nar/article/53/D1/D1099/7815638).

Additional relevant datasets that could further improve the benchmark include:
https://www.cell.com/cell-systems/fulltext/S2405-4712(24)00366-1
https://genomebiology.biomedcentral.com/articles/10.1186/s13059-024-03404-6
https://www.sciencedirect.com/science/article/pii/S0092867422005979?via%3Dihub
https://www.nature.com/articles/s41587-023-01964-9

Inclusion of such datasets would help improve coverage and generalizability of the benchmark.

**Strengths Contributions:**

The paper integrates datasets from multiple studies and offers a unified benchmark for perturbation prediction.

It re-implements several key baseline methods in a modular and transparent way, enabling easier reproducibility and ablation studies.

The proposed rank-based metric offers a potentially useful way to evaluate prediction quality in this context.

The findings challenge previous claims—for instance, showing that decoder-only models can perform well, possibly even without using control cell profiles, which is counterintuitive and worth further investigation.

Overall, the paper makes a strong case for the need for reproducibility and robust benchmarking in perturbation modeling.

---

> ### Author Rebuttal · Authors · 2025-07-31
>
> We would like to thank you for the insightful and positive review of our work, and for the helpful comments. We will address each comment below:
>
> > The claim that the rank-based metric has an expected value of 0.5 under random predictions is not empirically validated. It would strengthen the paper to include experiments using random or naive predictors to confirm this baseline behavior.
>
> Thank you for the suggestion - we ran 10 experiments using a null predictor (randomly initialized Latent Additive model) in Srivatsan20 dataset and achieved a rank metric of 0.4988 on RMSE average. We plan on adding these results to the manuscript.
>
> While running these experiments, we noticed a minor off by one error when computing the rank metrics, which made them slightly lower than they should have been. We were dividing by num_perturbations when it should have been num_perturbations - 1. This did not affect relative model performance at all and we will fix the error in the PerturBench repo.
>
> > The dataset is relatively limited in scope, covering only five publications. A more comprehensive benchmark would benefit from including broader datasets. For instance, PerturBase curates 122 datasets from 46 studies, covering over 24,000 genetic and 230 chemical perturbations from approximately 5 million cells. (see: https://academic.oup.com/nar/article/53/D1/D1099/7815638). [...] Inclusion of such datasets would help improve coverage and generalizability of the benchmark.
>
> Thank you for the suggestion to include additional datasets. To improve the generalizability of our benchmark, we plan to add the Open Problems Perturbation Prediction (OP3) benchmark dataset which includes chemical perturbations in primary human PBMCs, a more biologically realistic cell system than many of the immortalized cell lines used in other datasets [1].
>
> And thank you for bringing these other datasets to our attention. The in-vivo perturb-seq dataset from Liu et al looks especially interesting, and we also plan to also add it as a benchmark of how well these perturbation models can generalize from in-vitro to in-vivo settings. The multiome perturb-seq dataset from Metzner et al looks interesting as well, but at the moment most perturbation response models are not compatible with multimodal or multiomic datasets. We look forward to seeing more work in this space and hope to update our benchmark accordingly. The Replogle et al dataset is also interesting, but is more amenable to the unseen perturbation prediction task whereas our benchmark focuses more on the covariate transfer and combo prediction tasks.
>
> Given the diversity and scope of perturbational datasets with transcriptomic readouts, we recognized that it would be difficult to capture all datasets of interest and thus wanted to make our codebase easily extensible. We want to emphasize that adding a new dataset is as simple as running our preprocessing script and adding a data config file.
>
> [1] Szałata, Artur, et al. "A benchmark for prediction of transcriptomic responses to chemical perturbations across cell types." Advances in Neural Information Processing Systems 37 (2024): 20566-20616.

---

> > ### Author Response · Authors · 2025-08-05
> > **Additional dataset results**
> >
> > Dear reviewer jrGj,
> >
> > Thank you for acknowledging our rebuttal.
> >
> > We wanted to let you know that we have recently introduced the Open Problems Perturbation Prediction (OP3) dataset to our benchmark, which is from [1] and features chemical perturbations in primary human PBMCs. The performance of our models after hyperparameter optimization is below, split out into our original pseudobulk metrics and newly added distributional metrics including MMD and differentially expressed gene (DEG) recall.
> >
> > **OP3 Pseudobulk Metrics**
> > | model             | Cosine LogFC   | RMSE mean   | Cosine LogFC rank   | RMSE mean rank   |
> > |:------------------|:---------------|:---------------|:--------------------|:--------------------|
> > | CPA               | 0.42 ± 3E-03   | 0.034 ± 2E-04  | 0.071 ± 4E-03       | 0.048 ± 6E-03       |
> > | CPA noAdv         | 0.34 ± 8E-03   | 0.036 ± 5E-04  | 0.057 ± 4E-03       | 0.043 ± 3E-03       |
> > | SAMS-VAE          | 0.44 ± 1E-02   | 0.036 ± 7E-04  | 0.067 ± 2E-02       | 0.066 ± 2E-02       |
> > | SAMS-VAE improved | 0.46 ± 2E-03   | 0.033 ± 2E-04  | 0.053 ± 4E-03       | 0.040 ± 1E-03       |
> > | Latent     | 0.39 ± 2E-02   | 0.036 ± 2E-03  | 0.12 ± 3E-02        | 0.14 ± 3E-02        |
> > | Decoder           | 0.32 ± 6E-03   | 0.035 ± 2E-04  | 0.070 ± 5E-03       | 0.068 ± 6E-03       |
> > | Linear            | 0.011 ± 9E-05  | 0.070 ± 4E-05  | 0.14 ± 4E-04        | 0.24 ± 9E-04        |
> >
> > **OP3 MMD and DEG Metrics**
> > | model             | MMD GEX    | MMD PCA      | MMD GEX rank   | MMD PCA rank   | DEG Recall   |
> > |:------------------|:------------|:-------------|:----------------|:---------------|:----------------------|
> > | CPA               | 5.5 ± 3E-02 | 0.81 ± 9E-03 | 0.23 ± 2E-02    | 0.039 ± 8E-03  | 0.0044 ± 5E-04        |
> > | CPA noAdv         | 5.8 ± 4E-03 | 0.96 ± 2E-02 | 0.21 ± 1E-02    | 0.036 ± 5E-03  | 0 ± 0                 |
> > | SAMS-VAE          | 5.7 ± 4E-02 | 1.0 ± 4E-02  | 0.20 ± 8E-03    | 0.070 ± 1E-02  | 0 ± 0                 |
> > | SAMS-VAE improved | 5.6 ± 1E-02 | 0.83 ± 1E-02 | 0.23 ± 1E-02    | 0.044 ± 3E-03  | 0 ± 0                 |
> > | Latent     | 11. ± 6E-02 | 4.1 ± 5E-02  | 0.16 ± 2E-02    | 0.13 ± 3E-02   | 2.8e-05 ± 6E-05       |
> > | Decoder           | 11. ± 5E-03 | 4.4 ± 1E-02  | 0.077 ± 6E-03   | 0.055 ± 5E-03  | N/A        |
> > | Linear            | 4.0 ± 7E-03 | 2.6 ± 4E-03  | 0.12 ± 8E-04    | 0.25 ± 2E-03   | 0 ± 0                 |
> >
> > Could you please let us know if this update to our benchmark datasets has addressed your original concern about limited datasets? Thank you.
> >
> > [1] Szałata, Artur, et al. "A benchmark for prediction of transcriptomic responses to chemical perturbations across cell types." Advances in Neural Information Processing Systems 37 (2024): 20566-20616.

---

### Official Review · Reviewer_2oW4 · 2025-07-08

**Rating:** 5
**Confidence:** 2

**Summary:**

This paper introduces PerturBench, a comprehensive, modular benchmarking framework for evaluating machine learning models that predict cellular responses to genetic and chemical perturbations from single-cell transcriptomic data. The authors provide: 1)A curated set of datasets containing biologically relevant covariate transfer and combination tasks; 2) A customized set of evaluation metrics, including traditional metrics (e.g., RMSE, cosine similarity) and a newly proposed rank metric, which helps diagnose common failure modes such as mode collapse; 3) A flexible model development and evaluation codebase that supports ablation studies and scalability analysis; 4) A comprehensive experimental study on datasets of varying size and complexity, using both published and novel baseline models.

**Additional Feedback:**

No Additional Feedback.

**Dataset Code Accessibility:**

Yes

**Dataset Code Comments:**

The benchmark datasets are public and documented.

**Ethical Considerations:**

No, there are no or only very minor ethics concerns

**Limitations Weaknesses:**

I think the biggest issue with Limitations is the scope of the benchmark vs. model fidelity. While the benchmarking work is laudable, the authors note (Section 6) that the model implementation is only an approximation of the original paper. This may limit the generalizability of some of the conclusions about model performance. I think adding more transparent and quantitative analysis of the differences from the original implementation (e.g., hyperparameters, data preprocessing) would help strengthen the credibility of the model.

There are also some minor issues, such as the lack of cell-level evaluation. I found that all the metrics in the paper are population-level (mean, fold change), which may ignore the heterogeneity of perturbation effects at the single-cell level (Section 6, Limitations). Although distribution metrics are implemented, they are not core to the results. Future work could integrate and emphasize these metrics.

**Strengths Contributions:**

First, I think the important contribution lies in benchmarking. This paper provides a much-needed standardized method for evaluating perturbation prediction models. Incorporating real-world tasks (e.g., covariate transfer, combined prediction) and large-scale datasets enhances the relevance and usefulness of the model.
Second, novel evaluation metrics are also important, such as the ranking metric proposed in this paper, which directly addresses modal/posterior collapse, a key but under-diagnosed failure mode in generative models. This contribution is both novel and of practical significance.
Finally, there are comprehensive experiments and simplifications. Detailed benchmarks (Tables 2-4) show that simplified variants of some established models outperform the original versions, challenging the assumptions of previous studies. This study also evaluates data expansion (Figure 3), highlighting differences in model robustness.

---

> ### Author Rebuttal · Authors · 2025-07-31
>
> We would like to thank you for the insightful and positive review of our work, and for the helpful comments. We will address each comment below:
>
> > I think the biggest issue with Limitations is the scope of the benchmark vs. model fidelity. While the benchmarking work is laudable, the authors note (Section 6) that the model implementation is only an approximation of the original paper. This may limit the generalizability of some of the conclusions about model performance. I think adding more transparent and quantitative analysis of the differences from the original implementation (e.g., hyperparameters, data preprocessing) would help strengthen the credibility of the model.
>
> Thank you for bringing up the key issue of model fidelity. To partially address this issue and demonstrate independent usage of our evaluation module, we verified that our CPA implementation matches or exceeds the published CPA implementation performance on the Norman19 dataset, where CPA_Theis is the public version of CPA hosted on github and accessed on 07/30/2025, and CPA_perturbench is our recreation of CPA. We specifically trained a CPA_Theis model using their Norman19 example notebook with provided hyperparameters and their version of the Norman19 dataset, using the “split_6” train/val/test split. We then added their version of the Norman19 dataset to PerturBench, and trained our CPA implementation on the same dataset/split. We evaluated both model’s performance on the held out “ood” split using PerturBench metrics. We plan on adding these verification notebooks and config files to the PerturBench repo.
>
> |                         |   CPA_Theis |   CPA_perturbench |
> |:------------------------|------------:|-------------:|
> | rmse_average            |     0.1124  |     0.02729  |
> | rmse_rank_average       |     0.4267  |     0.008889 |
> | cosine_logfc            |     0.2058  |     0.7498   |
> | cosine_rank_logfc       |     0.2622  |     0.01333  |
> | r2_score_scores         |    -1.004   |     0.2247   |
> | top_k_recall_scores     |     0.09867 |     0.384    |
> | mmd_pca                 |     3.283   |     2.308    |
> | mmd_rank_pca            |     0.4222  |     0.008889 |
> | mmd_none                |     4.237   |     3.769    |
> | mmd_rank_none           |     0.4133  |     0.008889 |
>
> We noticed that the choice of loss has a major impact on performance, and models that used a simple MSE loss performed better than Gaussian or Negative Binomial Negative Log Likelihood (NLL) losses. We chose to use the MSE loss in our CPA reproduction, whereas the public model uses NLL losses we believe is the primary reason why our reimplemented CPA performs significantly better than the public CPA implementation. We feel that the use of MSE loss enables us to compare the rest of the model components more on a more even basis, but can add our findings on the NLL vs MSE loss to the manuscript as well.
>
> A detailed explanation of differences between our implementations and the public reference implementations are summarized in Appendix D.5. Please let us know if there are details that require further explanation. The users can also verify the details of implementation in the open source codebase that we released.
>
> Finally, the optimal hyperparameters depend on the dataset that is being trained on. We performed hyperparameter optimization for every model and every dataset, which should give us the best performance and a leveled playing field. Using hyperparameters from the original implementations would favour models developed on dataset similar to the ones in our benchmark.
>
> > There are also some minor issues, such as the lack of cell-level evaluation. I found that all the metrics in the paper are population-level (mean, fold change), which may ignore the heterogeneity of perturbation effects at the single-cell level (Section 6, Limitations). Although distribution metrics are implemented, they are not core to the results. Future work could integrate and emphasize these metrics.
>
> Thank you for the comment on cell level metrics. We have added results for the maximum mean discrepancy (MMD) and differentially expressed gene (DEG) recall metrics for the Jiang24, Norman19 and Srivatsan20 datasets. To briefly introduce these metrics:
> MMD computes the energy distance between predicted cells and the ground truth perturbed cells, either in the gene expression (MMD GEX) or PCA (MMD PCA) space. We use the geomloss implementation with the energy kernel and default parameters. To compute PCs, we fit a PCA model to the ground truth test split, and project predicted cells onto the ground truth PCs. We also compute the corresponding rank metrics, MMD GEX rank and MMD PCA rank.
> DEG recall measures the fraction of ground truth DEGs that can be recalled from the predicted cells. We computed t-scores using the scanpy.tl.rank_genes_groups method, and compared them using top k recall. Since the DecoderOnly model has a variance of zero, we set its DEG recall to N/A since it is difficult to compare to the other models.
>
> **Jiang24**
> | model             | MMD GEX     | MMD PCA      | MMD GEX rank   | MMD PCA rank   | DEG recall      |
> |:------------------|:------------|:-------------|:---------------|:---------------|:----------------|
> | CPA               | 8.0 ± 2E-03 | 2.5 ± 4E-03  | 0.44 ± 7E-03   | 0.43 ± 4E-03   | 0.0044 ± 2E-03  |
> | CPA noAdv         | 8.0 ± 2E-03 | 2.5 ± 3E-03  | 0.44 ± 6E-03   | 0.42 ± 4E-03   | 0.0052 ± 6E-04  |
> | SAMS-VAE          | 4.5 ± 7E-03 | 0.32 ± 5E-03 | 0.47 ± 3E-03   | 0.43 ± 1E-02   | 1.8e-05 ± 4E-05 |
> | SAMS-VAE improved | 6.5 ± 2E+00 | 1.6 ± 1E+00  | 0.44 ± 2E-02   | 0.43 ± 1E-02   | 0.0015 ± 1E-04  |
> | Latent            | 8.0 ± 9E-04 | 2.6 ± 2E-03  | 0.41 ± 5E-03   | 0.40 ± 5E-03   | 0.0010 ± 5E-04  |
> | Decoder           | 8.0 ± 2E-03 | 2.6 ± 2E-03  | 0.36 ± 9E-03   | 0.38 ± 8E-03   | N/A   |
> | Linear            | 3.1 ± 2E-03 | 1.3 ± 2E-03  | 0.44 ± 1E-03   | 0.45 ± 6E-04   | 0.0033 ± 2E-04  |
>
>
> **Norman19**
> | model             | MMD GEX     | MMD PCA      | MMD GEX rank   | MMD PCA rank   | DEG recall      |
> |:------------------|:------------|:-------------|:---------------|:---------------|:----------------|
> | CPA               | 5.6 ± 3E-02 | 2.2 ± 2E-02  | 0.14 ± 9E-03   | 0.044 ± 5E-03  | 0.032 ± 4E-03   |
> | CPA noAdv         | 5.5 ± 1E-01 | 2.2 ± 1E-01  | 0.13 ± 2E-02   | 0.038 ± 5E-03  | 0.016 ± 9E-03   |
> | SAMS-VAE          | 4.1 ± 4E-02 | 1.9 ± 3E-02  | 0.034 ± 1E-02  | 0.020 ± 2E-03  | 0 ± 0           |
> | SAMS-VAE improved | 3.3 ± 5E-02 | 0.74 ± 5E-02 | 0.088 ± 6E-03  | 0.029 ± 4E-03  | 0.028 ± 6E-03   |
> | Biolord           | 2.8 ± 5E-03 | 1.6 ± 5E-03  | 0.066 ± 3E-03  | 0.042 ± 3E-03  | 0 ± 0           |
> | Latent            | 6.7 ± 4E-03 | 3.2 ± 6E-03  | 0.012 ± 1E-03  | 0.013 ± 8E-04  | 0.00043 ± 3E-04 |
> | Decoder           | 6.7 ± 6E-03 | 3.2 ± 4E-03  | 0.0090 ± 8E-04 | 0.0098 ± 4E-04 | N/A    |
> | Linear            | 2.5 ± 4E-02 | 1.2 ± 4E-02  | 0.026 ± 2E-03  | 0.017 ± 4E-04  | 0.018 ± 2E-03   |
>
>
>
> **Srivatsan20**
> | model             | MMD GEX     | MMD PCA      | MMD GEX rank   | MMD PCA rank   | DEG recall      |
> |:------------------|:------------|:-------------|:---------------|:---------------|:----------------|
> | CPA               | 2.4 ± 1E-02 | 0.53 ± 4E-03 | 0.30 ± 9E-03   | 0.20 ± 9E-03   | 0.0073 ± 2E-03  |
> | CPA noAdv         | 2.3 ± 3E-02 | 0.49 ± 1E-02 | 0.25 ± 5E-03   | 0.13 ± 1E-02   | 0.0040 ± 2E-03  |
> | SAMS-VAE          | 2.5 ± 2E-02 | 0.69 ± 1E-02 | 0.30 ± 8E-03   | 0.060 ± 3E-03  | 0.00016 ± 1E-04 |
> | SAMS-VAE improved | 2.9 ± 1E-02 | 0.79 ± 1E-02 | 0.28 ± 5E-03   | 0.18 ± 2E-02   | 0 ± 0           |
> | Biolord           | 4.9 ± 3E+00 | 4.3 ± 4E+00  | 0.36 ± 2E-01   | 0.32 ± 2E-01   | 4.7e-05 ± 1E-04 |
> | Latent            | 4.3 ± 2E-01 | 2.0 ± 2E-01  | 0.26 ± 6E-02   | 0.26 ± 5E-02   | 0.0 ± 0.0       |
> | Decoder           | 4.2 ± 4E-03 | 1.9 ± 5E-03  | 0.16 ± 2E-02   | 0.15 ± 1E-02   | N/A  |
> | Linear            | 2.2 ± 7E-03 | 0.76 ± 9E-04 | 0.31 ± 1E-03   | 0.30 ± 3E-04   | 0.0036 ± 3E-04  |
>
> The MMD and DEG recall metrics assess the full distribution of cellular responses to perturbation. The SAMS-VAE and CPA reimplementations perform better on MMD PCA than the baseline Latent and Decoder models, while the baseline models still outperform the published models on the MMD PCA rank metrics.
>
> All models performed poorly on DEG recall, suggesting that they struggle to predict differential expression t-scores. Models also performed worse on MMD GEX compared to MMD PCA, suggesting that predicting gene expression distributions is still a major challenge. SAMS-VAE and CPA still outperformed the baseline Latent and Decoder models for MMD GEX.
>
> We plan to add the MMD and DEG recall metrics for all datasets to the manuscript.

---

### Decision · Program_Chairs · 2025-09-18

**Decision:**

Accept (poster)

**Comment:**

This paper presents PerturBench a framework for perturbation response modeling. The contribution lies in curating several Perturb-seq datasets and providing standardizing tasks and introducing new rank-based metrics. The reviewers agreed it is an important contribution as perturbation is imporotant to drug discovery and genomics. The experiments are thorough and reveal meaningful insights. Overall I find this to be a valuable resource for the community and recommend it is accepted.